# A symbolic framework to obtain mid-fidelity models of flexible multibody systems with application to horizontal-axis wind turbines

Emmanuel Branlard[1] and Jens Geisler [2]

[1]National Renewable Energy Laboratory, Golden, CO 80401, USA
[2]Hochschule Flensburg, University of Applied Sciences, 24943 Flensburg, Germany

**Correspondence:** E. Branlard (emmanuel.branlard@nrel.gov)

**Abstract.** The article presents a symbolic framework (also called computer algebra program) that is used to obtain, in symbolic
mathematical form, the linear and nonlinear equations of motion of a mid-fidelity multibody system including rigid and flexible
bodies . Our approach is based on Kane's method and a nonlinear shape function representation for flexible bodies. The shape
function approach does not represent the state of the art for flexible multibody dynamics but it represents an effective trade-off to
obtain mid-fidelity models with few degrees of freedom and taking advantage of the separation of space and time. The method
yields compact symbolic equations of motion with implicit account of the constraints. The general and automatic framework
facilitates the creation and manipulation of models with various levels of complexity by adding or removing degrees of freedom.
The symbolic treatment allows for analytical gradients and linearized equations of motion. The linear and nonlinear equations
can be exported to Python code or dedicated software. There are multiple applications, such as time domain simulation, stability
analyses, frequency domain analyses, advanced controller design, state observers, and digital twins. In this article, we describe
the method we used to systematically generate the equations of motion of multibody systems, and present the implementation
of the framework using the Python package SymPy. We apply the framework to generate illustrative land-based and offshore
wind turbine models. We compare our results with OpenFAST simulations and discuss the advantages and limitations of the
method. The Python implementation is provided as an open-source project.

## 1   Introduction

The next generation of wind turbine digital technologies and control systems require versatile aero-servo-hydro-elastic models,
with various levels of fidelity, suitable for a wide range of applications. Such applications include time domain simulations, lin-
earization (for controller design and tuning, or frequency domain analyses), analytical gradients (for optimization procedures),
and generation of dedicated, high-performance or embedded code (for stand-alone simulations, state observers or digital twins).
Current models are implemented for a specific purpose and are usually based on an heuristic structure. Aeroelastic tools, such
as Flex (Øye, 1983; Branlard, 2019) or ElastoDyn (Jonkman et al., 2021), rely on an assumed chain of connections between
bodies, a given set of degrees of freedom, and predefined orientations of shape functions. It is not straightforward to extract
reduced-order models from these tools or extend the models to additional degrees of freedom.

Tools with linearization capabilities, such as HAWCStab2 (Sønderby and Hansen, 2014) or OpenFAST (Jonkman et al., 2021) are dedicated to horizontal-axis wind turbines, and the evaluation of the gradients are limited to hard-coded analytical expressions or numerical finite differences. Small implementation changes often require extensive redevelopment, and the range of applications of the tools remains limited (Simani, 2015). The linear models generated by these tools are numerical models that are evaluated for a given set of numerical input parameters. It is therefore difficult to obtain gradients of the linear models as function of the input parameters, an information which is becoming increasingly important in optimization frameworks and controls co-design approaches (Jonkman et al., 2022).

To address these issues, we propose a symbolic framework (also called a computer algebra program) for the automatic derivation, processing, and parameterization of models with granularity in the level of fidelity. Our approach is based on Kane's method (Kane and Wang, 1965) and a nonlinear shape function representation of flexible bodies (Shabana, 2013) described using a standard input data (SID) format (Wallrapp, 1994; Schwertassek and Wallrapp, 1999). The method yields compact symbolic equations of motion with implicit account of the constraints. Similar approaches have been presented in the literature: Kurz and Eberhard (2009), Merz (2018), Lemmer (2018), and Branlard (2019). Our framework differs in the fact that all equations are processed at a symbolic level and therefore the model can be used in its nonlinear or linearized form. The linear models are obtained using analytical differentiation. They can be be evaluated for various set of input parameters directly and therefore be used in optimization frameworks or controls co-design approaches. We implemented an open-source version in Python using SymPy (SymPy, 2021), leveraging its mechanical toolbox. Alternative symbolic frameworks found in the literature are usually limited to rigid bodies (Verlinden et al., 2005; Kurz and Eberhard, 2009; Gede et al., 2013; Docquier et al., 2013) or are closed-source or using proprietary software (Reckdahl and Mitiguy, 1996; Kurtz et al., 2010; MotionGenesis, 2016; Lemmer, 2018).

Kane's method and the nonlinear shape function approach presented in this article do not represent the state of the art of multibody dynamics with flexible bodies. The geometrically exact beam theory (Simo, 1985; Jelenić and Crisfield, 1999; Géradin and Cardona, 2001; Bauchau, 2011) is more precise than the shape function approach because it represents the beam kinematics exactly. Linearization of the geometrical exact beam theory equations is possible and also more precise than the shape function approach but it leads to larger and more involved expressions. Similarly, multipurpose multibody software exists (Lange et al., 2007), such as ANSYS (ANSYS, 2022), SIMPACK (SIMPACK, 2022), or MBDyn (MBDyn, 2022). These more advanced approaches target different applications than those envisioned in this study: they are suitable for numerical simulations, but they cannot provide symbolic mid-fidelity models in compact form.

In section 2, we present the formalism used to derive the equations of motion in a systematic and unified way for flexible and rigid bodies. In section 3, we give an overview of how the symbolic calculation framework was implemented. Example of applications relevant to wind energy are given in section 4. Discussions and conclusions follow.

## 2 Method to obtain the equations of motion

In this section, we present the formalism used to setup the equations of motion.

## 2.1 System definition and kinematics

We consider a system of $n_b$ bodies, rigid or flexible, connected by a set of joints. For simplicity, we assume that no kinematic loops are present in the system, and the masses of the bodies are constant. An inertial frame is defined to express the positions, velocities, and accelerations of the bodies. We adopt a minimal set of generalized coordinates, $q$, of dimension $n_q$, to describe the kinematics of the bodies: joint coordinates describing the joints displacements, and Rayleigh-Ritz coordinates for the amplitudes of the shape functions of the flexible bodies (see, e.g., Branlard (2019)). The choice of coordinates is left to the user, but it is assumed to form a minimal set. We will provide illustrative examples in section 4.

At a given time, the positions, orientations, velocities, and accelerations of all the points of the structure are entirely determined by the knowledge of $q, \dot{q}$, and $\ddot{q}$, where $(\dot{\cdot})$ represents the time derivative. For a given body $i$, and a point $P$ belonging to the body, the position, velocity, and acceleration of the point are given by (see, e.g., Shabana (2013)):

$$r_P = r_i + s_P = r_i + s_{P_0} + u_P \tag{1}$$

$$v_P = v_i + \omega_i \times s_P + (\dot{u}_P)_i \tag{2}$$

$$a_P = a_i + \omega_i \times (\omega_i \times s_P) + \dot{\omega}_i \times s_P + 2\omega_i \times (\dot{u}_P)_i + (\ddot{u}_P)_i \tag{3}$$

where $r_i$, $v_i$, and $a_i$ are the position, velocity, and acceleration of the origin of the body, respectively; $s_{P_0}$ is the initial (undeformed) position vector of point $P$ with respect to the body origin; the subscript $P$ is used for the deformed position of the point and $P_0$ for the underformed position; $u_P$ is the elastic displacement of the point (equal to 0 for rigid bodies); $\omega_i$ is the rotational velocity of the body with respect to the inertial frame; $(\dot{\cdot})$ and $(\dot{\cdot})_i$ refer to time derivatives in the inertial and body frame respectively. Throughout the article, we use bold symbols for vectors and matrices, and uppercase symbols for most matrices. The elastic displacement is obtained as a superposition of elastic deformations (see subsection 2.4). We define the transformation matrix $R_i$ that transforms coordinates from the body frame to the inertial frame, and by definition $[\tilde{\omega}_i] = \dot{R}_i R_i^T$, where $[\tilde{\cdot}]$ represents the skew symmetric matrix, and the exponent $T$ denotes the matrix transpose. We assume that vectors are represented as column vectors to conveniently introduce matrix-vector multiplications. We use the notation "$\cdot$" to indicate the dot product between two vectors (irrespective of their column or row representation).

## 2.2 Introduction to Kane's method

Kane's method (Kane and Wang, 1965) is a powerful and systematic way to obtain the equations of motion of a system. The procedure leads to $n_q$ coupled equations of motion:

$$\mathrm{f}_r + \mathrm{f}_r^* = 0, \qquad r = 1 \dots n_q \tag{4}$$

where $\mathrm{f}_r^*$ is associated with inertial loads and $\mathrm{f}_r$ is associated with external loads, and these components are obtained for all generalized coordinates. The components are obtained as a superposition of contributions from each body:

$$\mathrm{f}_r = \sum_{i=1}^{n_b} \mathrm{f}_{ri}, \qquad \mathrm{f}_r^* = \sum_{i=1}^{n_b} \mathrm{f}_{ri}^* \tag{5}$$

The terms $f_{ri}$ and $f_{ri}^*$ can be obtained for each body individually and assembled at the end to form the final system of equa-
tions. We will present in subsection 2.3 and subsection 2.4 how these terms are defined for rigid bodies and flexible bodies,
respectively.

## 2.3  Rigid bodies

We assume that body $i$ is a rigid body and proceed to define the terms $f_{ri}$ and $f_{ri}^*$. The inertial force, $\boldsymbol{f}_i^*$, and inertial torque,
$\boldsymbol{\tau}_i^*$, acting on the body are:

$$\boldsymbol{f}_i^* = -m_i \boldsymbol{a}_{G,i}, \qquad \boldsymbol{\tau}_i^* = -\boldsymbol{I}_{G,i} \cdot \dot{\boldsymbol{\omega}}_i - \boldsymbol{\omega}_i \times (\boldsymbol{I}_{G,i} \cdot \boldsymbol{\omega}_i) \tag{6}$$

where $m_i$ is the mass of the body, $\boldsymbol{a}_{G,i}$ is the acceleration of its center of mass with respect to the inertial frame, and $\boldsymbol{I}_{G,i}$ is the
inertial tensor of the body expressed at its center of mass. Equation 6 is a vectorial relationship; it may therefore be evaluated
in any coordinate system. The component $f_{ri}^*$ is defined as:

$$f_{ri}^* = \boldsymbol{J}_{v,ri} \cdot \boldsymbol{f}_i^* + \boldsymbol{J}_{\omega,ri} \cdot \boldsymbol{\tau}_i^* \tag{7}$$

with

$$\boldsymbol{J}_{v,ri} = \frac{\partial \boldsymbol{v}_{G,i}}{\partial \dot{q}_r}, \quad \boldsymbol{J}_{\omega,ri} = \frac{\partial \boldsymbol{\omega}_i}{\partial \dot{q}_r} \tag{8}$$

where $\boldsymbol{v}_{G,i}$ is the velocity of the body mass center with respect to the inertial frame. The partial velocities, or Jacobians, $\boldsymbol{J}_v$
and $\boldsymbol{J}_\omega$, are key variables of the Kane's method. They project the physical coordinates into the generalized coordinates ($\boldsymbol{q}$),
inherently accounting for the kinematic constraints between bodies. In numerical implementations, the Jacobians are typically
stored in matricial forms, referred to as "velocity transformation matrices." The terms $f_{ri}^*$ can equivalently be obtained using the
partial velocity of any body point (e.g., the origin) by carefully transferring the inertial loads to the chosen point. The external
forces and torques acting on the body are combined into an equivalent force and torque acting at the center of mass, written as
$\boldsymbol{f}_i$ and $\boldsymbol{\tau}_i$. The component $f_{ri}$ is then given by:

$$f_{ri} = \boldsymbol{J}_{v,ri} \cdot \boldsymbol{f}_i + \boldsymbol{J}_{\omega,ri} \cdot \boldsymbol{\tau}_i \tag{9}$$

Equivalently, the contributions from each individual force, $\boldsymbol{f}_{i,j}$, acting on a point $P_j$ of the body $i$, and each torque, $\boldsymbol{\tau}_{i,k}$, can
be summed using the appropriate partial velocity to obtain $f_{ri}$:

$$f_{ri} = \sum_j \frac{\partial \boldsymbol{v}_{P_j}}{\partial \dot{q}_r} \cdot \boldsymbol{f}_{i,j} + \sum_k \boldsymbol{J}_{\omega,ri} \cdot \boldsymbol{\tau}_{i,k} \tag{10}$$

where $\boldsymbol{v}_{P_j}$ is the velocity of the point $j$ with respect to the inertial frame. Equation 7 and Equation 9 are inserted into Equation 5
to obtain the final equations of motion.

## 2.4 Flexible bodies

We assume that body $i$ is a flexible body and proceed to define the terms $\mathrm{f}_{ri}$ and $\mathrm{f}_{ri}^*$. The dynamics of a flexible body are described in standards textbooks such as Shabana (2013) or Schwertassek and Wallrapp (1999). Unlike rigid bodies, the equations for flexible bodies are typically expressed with respect to a reference point different from the center of mass. We will call this point the origin and write it $O_i$. The elastic displacement field of the body is written as $\boldsymbol{u}$. It defines the displacement of any point of the body with respect to its undeformed position. Using the zeroth-order[1] Rayleigh-Ritz approximation, the displacement field at a given point, $P$, is given by the sum of shape function contributions: $\boldsymbol{u}(P) = \sum_{j=1}^{n_{e,i}} \boldsymbol{\Phi}_{ij}(P)\boldsymbol{q}_{e,ij}(t)$, where $\boldsymbol{\Phi}_{ij}$ are the shape functions (displacement fields) of body $i$, and $\boldsymbol{q}_{e,ij}$ is the subset of $\boldsymbol{q}$ consisting of the elastic coordinates of body $i$, of size $n_{e,i}$. The principles of the shape function approach applied to beams are given in Appendix B. The shape functions are more easily represented in the body coordinate system. Vectors and matrices that are explicitly written in the body frame will be written with primes. The equations of motion of the flexible bodies are (Wallrapp, 1994):

$$
\begin{bmatrix} \boldsymbol{M}'_{xx} & \boldsymbol{M}'_{x\theta} & \boldsymbol{M}'_{xe} \\ & \boldsymbol{M}'_{\theta\theta} & \boldsymbol{M}'_{\theta e} \\ \text{sym.} & & \boldsymbol{M}'_{ee} \end{bmatrix}_i \begin{bmatrix} \boldsymbol{a}'_i \\ \dot{\boldsymbol{\omega}}'_i \\ \ddot{\boldsymbol{q}}_{e,i} \end{bmatrix} + \begin{bmatrix} \boldsymbol{k}'_{\omega,x} \\ \boldsymbol{k}'_{\omega,\theta} \\ \boldsymbol{k}'_{\omega,e} \end{bmatrix}_i + \begin{bmatrix} 0 \\ 0 \\ \boldsymbol{k}_e \end{bmatrix}_i = \begin{bmatrix} \boldsymbol{f}'_x \\ \boldsymbol{f}'_\theta \\ \boldsymbol{f}_e \end{bmatrix}_i \tag{11}
$$

where the $x$, $\theta$, and $e$, subscripts respectively indicate the translation, rotation, and elastic components; $\boldsymbol{M}$ is the mass matrix of dimension $6+n_{e,i}$ made of the block matrices $\boldsymbol{M}_{xx}, \cdots, \boldsymbol{M}_{ee}$; $\boldsymbol{a}_i$ and $\dot{\boldsymbol{\omega}}_i$ are the linear and angular acceleration of the body (origin) with respect to the inertial frame; $\boldsymbol{k}_\omega$ are the centrifugal, gyration, and Coriolis loads, also called quadratic velocity loads; $\boldsymbol{k}_e$ are the elastic strain loads, which may contain geometric stiffening effects; $\boldsymbol{f}$ are the external forces, torques, and elastic generalized forces. The different components of $\boldsymbol{M}$, $\boldsymbol{k}_\omega$, $\boldsymbol{k}_e$, and $\boldsymbol{f}$ are given in Appendix A. These terms depend on $\boldsymbol{q}$, $\dot{\boldsymbol{q}}$, and $\boldsymbol{\Phi}_i$. The inertial force, torque, and elastic loads are:

$$\boldsymbol{f}_i^* = -\boldsymbol{R}_i \left[ \boldsymbol{M}'_{xx} \boldsymbol{a}'_i + \boldsymbol{M}'_{x\theta} \dot{\boldsymbol{\omega}}'_i + \boldsymbol{M}_{xe} \ddot{\boldsymbol{q}}_{e,i} + \boldsymbol{k}'_{\omega,x} \right] \tag{12}$$

$$\boldsymbol{\tau}_i^* = -\boldsymbol{R}_i \left[ \boldsymbol{M}'_{\theta x} \boldsymbol{a}'_i + \boldsymbol{M}'_{\theta\theta} \dot{\boldsymbol{\omega}}'_i + \boldsymbol{M}_{\theta e} \ddot{\boldsymbol{q}}_{e,i} + \boldsymbol{k}'_{\omega,\theta} \right] \tag{13}$$

$$\boldsymbol{h}_i^* = - \quad \left[ \boldsymbol{M}'_{ex} \boldsymbol{a}'_i + \boldsymbol{M}'_{e\theta} \dot{\boldsymbol{\omega}}'_i + \boldsymbol{M}_{ee} \ddot{\boldsymbol{q}}_{e,i} + \boldsymbol{k}'_{\omega,e} \right] \tag{14}$$

The external and elastic loads are:

$$\boldsymbol{f}_i = \boldsymbol{R}_i \boldsymbol{f}'_x \tag{15}$$

$$\boldsymbol{\tau}_i = \boldsymbol{R}_i \boldsymbol{f}'_\theta \tag{16}$$

$$\boldsymbol{h}_i = \boldsymbol{f}_e - \boldsymbol{k}_e \tag{17}$$

The components of $\mathrm{f}_{ri}^*$ and $\mathrm{f}_{ri}$, for $r = 1 \cdots n_q$, are then defined as:

$$\mathrm{f}_{ri}^* = \boldsymbol{J}_{v,ri} \cdot \boldsymbol{f}_i^* + \boldsymbol{J}_{\omega,ri} \cdot \boldsymbol{\tau}_i^* + \boldsymbol{J}_{e,ri} \cdot \boldsymbol{h}_i^* \tag{18}$$

$$\mathrm{f}_{ri} = \boldsymbol{J}_{v,ri} \cdot \boldsymbol{f}_i + \boldsymbol{J}_{\omega,ri} \cdot \boldsymbol{\tau}_i + \boldsymbol{J}_{e,ri} \cdot \boldsymbol{h}_i \tag{19}$$

---

[1]We address the first-order approximation in Appendix D4.

with
$$\boldsymbol{J}_{v,ri} = \frac{\partial \boldsymbol{v}_{O,i}}{\partial \dot{q}_r}, \quad \boldsymbol{J}_{\omega,ri} = \frac{\partial \boldsymbol{\omega}_i}{\partial \dot{q}_r}, \quad \boldsymbol{J}_{e,ri} = \frac{\partial \boldsymbol{q}_{e,i}}{\partial q_r} \tag{20}$$
where $\boldsymbol{v}_{O,i}$ is the velocity of the body with respect to the inertial frame. The term $\boldsymbol{J}_{e,ri}$ consists of 0 and 1 because $\boldsymbol{q}_{e,i}$ is a
subset of $\boldsymbol{q}$. Equation 18 and Equation 19, once evaluated for body $i$, are inserted into Equation 5 to obtain the final equations
of motion.

## 2.5  Nonlinear and linear equations of motion

The $n_q$ equations of motion given in Equation 4 are gathered into a vertical vector $\mathbf{e}$. They are recast into the form:
$$\mathbf{e}(\boldsymbol{q}, \dot{\boldsymbol{q}}, \ddot{\boldsymbol{q}}, \boldsymbol{u}, t) = \mathbf{f} + \mathbf{f}^* = \boldsymbol{F}(\boldsymbol{q}, \dot{\boldsymbol{q}}, \boldsymbol{u}, t) - \boldsymbol{M}(\boldsymbol{q})\ddot{\boldsymbol{q}} = \mathbf{0} \tag{21}$$
or
$$\boldsymbol{M}(\boldsymbol{q})\ddot{\boldsymbol{q}} = \boldsymbol{F}(\boldsymbol{q}, \dot{\boldsymbol{q}}, \boldsymbol{u}, t) \tag{22}$$
where $\boldsymbol{M} = -\frac{\partial \mathbf{e}}{\partial \ddot{\boldsymbol{q}}}$ is the system mass matrix and $\boldsymbol{F}$ is the forcing term vector—that is, the remainder terms of the equation
($\boldsymbol{F} = \mathbf{e} + \boldsymbol{M}\ddot{\boldsymbol{q}}$). The vector $\boldsymbol{u}$ is introduced to represent the time-dependent inputs that are involved in the determination of the
external loads. Both sides of the equations are also dependent on some parameters, but this dependency is omitted to shorten
notations. The stiffness and damping matrices may be obtained by computing the Jacobian of the equations of motion with
respect to $\boldsymbol{q}$ and $\dot{\boldsymbol{q}}$, respectively. The nonlinear equation given in Equation 22 is easily integrated numerically, for instance by
recasting the system into a first-order system, or by using a dedicated second-order system time integrator.
In various applications, a linear time invariant approximation of the system is desired. Such approximation is obtained at an
operating point, noted with the subscript 0, which is a solution of the nonlinear equations of motion, namely:
$$\mathbf{e}(\boldsymbol{q}_0, \dot{\boldsymbol{q}}_0, \ddot{\boldsymbol{q}}_0, \boldsymbol{u}_0, t) = \mathbf{0} \tag{23}$$
The linearized equations about this operating point are obtained using a Taylor series expansion:
$$\boldsymbol{M}_0(\boldsymbol{q}_0)\boldsymbol{\delta}\ddot{\boldsymbol{q}} + \boldsymbol{C}_0(\boldsymbol{q}_0, \dot{\boldsymbol{q}}_0, \boldsymbol{u}_0)\boldsymbol{\delta}\dot{\boldsymbol{q}} + \boldsymbol{K}_0(\boldsymbol{q}_0, \dot{\boldsymbol{q}}_0, \ddot{\boldsymbol{q}}_0, \boldsymbol{u}_0)\boldsymbol{\delta}\boldsymbol{q} = \boldsymbol{Q}_0(\boldsymbol{q}_0, \dot{\boldsymbol{q}}_0, \boldsymbol{u}_0)\boldsymbol{\delta}\boldsymbol{u} \tag{24}$$
with
$$\boldsymbol{M}_0 = -\left.\frac{\partial \mathbf{e}}{\partial \ddot{\boldsymbol{q}}}\right|_0, \; \boldsymbol{C}_0 = -\left.\frac{\partial \mathbf{e}}{\partial \dot{\boldsymbol{q}}}\right|_0, \; \boldsymbol{K}_0 = -\left.\frac{\partial \mathbf{e}}{\partial \boldsymbol{q}}\right|_0, \; \boldsymbol{Q}_0 = \left.\frac{\partial \mathbf{e}}{\partial \boldsymbol{u}}\right|_0 \tag{25}$$
where $\boldsymbol{M}_0$, $\boldsymbol{C}_0$, and $\boldsymbol{K}_0$ are the linear mass, damping, and stiffness matrices, respectively; $\boldsymbol{Q}_0\boldsymbol{\delta}\boldsymbol{u}$ is the linear forcing vector
($\boldsymbol{Q}_0$ is the input matrix); $\delta$ indicates a small perturbation of the quantities; and $|_0$ indicates that the expressions are evaluated at
the operating point. In practical applications, linearization is done at an operating point where the acceleration is zero ($\ddot{\boldsymbol{q}}_0 = 0$)
and most velocities are also zero. Examples of applications of the linear equations of motion are controller design, frequency
domain analyses, and stability analyses. The symbolic system matrices also allow for the easy formulation of linear parameter-
varying models used in many advanced control applications.

## 3 Implementation into a symbolic framework

In this section, we discuss the Python open-source symbolic calculation framework that we implemented according to the equations given in section 2. A Maxima implementation from the same authors is also available (Geisler, 2021).

The Python library YAMS (Yet Another Multibody Solver) started as a numerical tool published in previous work (Branlard, 2019). The library is now supplemented with a symbolic module so that both numerical and symbolic calculations can be achieved. The new implementation uses the Python symbolic calculation package SymPy (SymPy, 2021). We leveraged the features present in the subpackage "mechanics," which contains all the tools necessary to compute kinematics: the definition of frames and points and the determination of positions, velocities, and accelerations. The subpackage also contains an implementation of Kane's equations for rigid bodies (i.e., subsection 2.3). We were also inspired by the package PyDy (Gede et al., 2013), which is a convenient tool to export the equations of motion to executable code and directly visualize the bodies in 3D. The core of our work consisted of implementing a class to define flexible bodies (`FlexibleBody`) and the corresponding Kane's method for this class (subsection 2.4).

For the `FlexibleBody` class, we followed the formalism of Wallrapp (1994) and implemented Taylor expansions for all the terms defined in Appendix A, allowing the symbolic computation with Taylor expansions to any order. In practice, a zeroth- or first-order expansion is used. The use of Taylor expansions is presented in Appendix D3. The different Taylor coefficients may be kept as symbolic terms, or replaced early on by numerical values provided by a SID, for instance.

We structured the code into three layers: 1) The low-level layer integrates seamlessly with SymPy and PyDy by using the `FlexibleBody` class we provide. It is the layer that offers the highest level of granularity and control for the user, since arbitrary systems with various kinematic constraints can be implemented, at the cost of requiring more expertise. 2) The second-level automates the calculation of the kinematics by introducing simple connections between rigid and flexible bodies. The connections may be rigid, with constant offsets and rotations, or dynamic. A connection from a flexible body to another body is assumed to occur at one extremity of the flexible body. Some knowledge of SymPy mechanics is still required to use this layer. 3) The third level consists of template models such as generic land-based or offshore wind turbine models. Degrees of freedom are easily turned on and off for these conceptual models depending on the level of fidelity asked by the user, and generic external forces can be implemented or declared as external inputs.

The overall workflow for typical usage of the symbolic framework is illustrated in Figure 1. The symbolic framework takes as input a conceptual model of the structure, which is assembled using one of the three layers previously described. The nonlinear and linear equations of motion can be exported to LaTeX and Python-ready scripts for various applications (see subsection 5.1). Using the third layer, as little as three lines of code are required by the user to perform the full step from derivation of the equations, optional linearization, and exportation. To obtain numerical results from the exported Python code, the user needs to provide the arrays with the degrees of freedom values $q$ and $\dot{q}$, their initial conditions, a dictionary with inputs ($u$) that are functions of time, and a dictionary of parameters ($p$) containing all the numerical constants such as mass, acceleration of gravity, and geometric parameters. We implemented various preprocessing tools in YAMS to facilitate the calculation of numerical parameters, typically from a set of OpenFAST input files or by using structural parameters defined

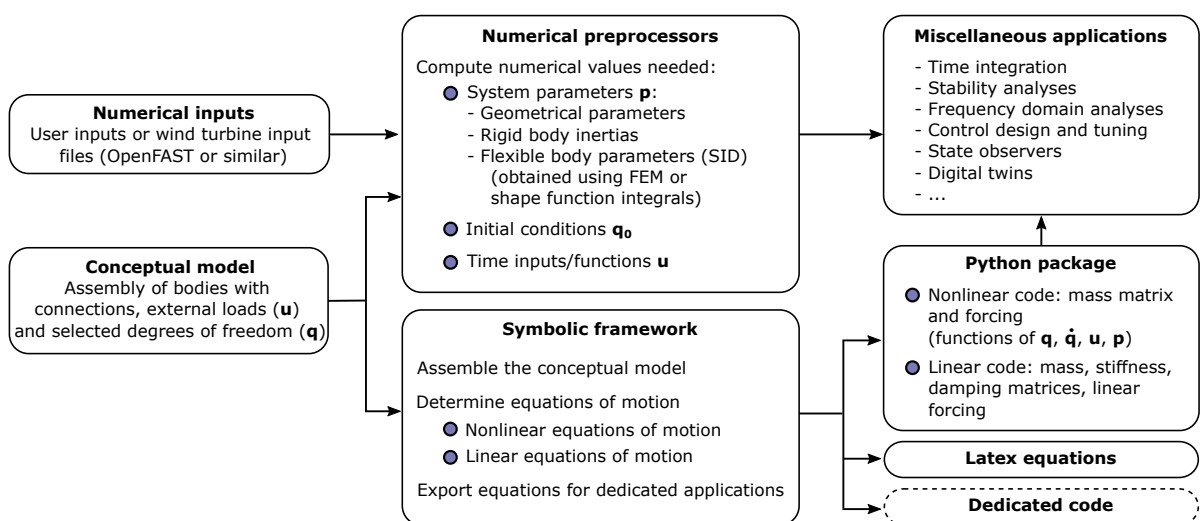

**Figure 1.** Typical workflow for the usage of the symbolic framework, going from numerical inputs and a conceptual model to numerical packages that can be used for various applications.

by the users. YAMS contains tools to compute the flexible bodies parameters (mass matrix, stiffness matrix, shape integrals)
using integrals over the shape functions or using a finite-element beam formulation. YAMS also contains tools to compute the
rigid body inertia of different components of a wind turbine or the full system. Postprocessing tools are also included to readily
time-integrate the generated model using numerical values (including initial values).
The source code of YAMS is available on GitHub as a subpackage of the Wind Energy LIBrary, WELIB (Branlard, 2021).
The repository contains tests and working examples, including the ones presented in section 4.

## 210   4   Wind energy applications

### 211   4.1   Approach

In this section, we present different wind energy applications of the symbolic framework. We focus on models with at least
one flexible body because the rigid body formulation of SymPy has been well verified (Gede et al., 2013). For each example,
the equations of motion are given and their results are compared with OpenFAST (Jonkman et al., 2021) simulations. This
is readily achieved because our framework can export the equations of motion to Python functions, load input files from an
OpenFAST model, and integrate the generated equations using the same conditions as defined in the OpenFAST input files.
In this article, we do not focus on the modeling of the external loads, but we include them in the equations of motion. It is
the responsibility of the user to define these functions, for instance through aero- or hydro-force models. For the verification
results presented in this section, we only include the gravitational and inertial loading. In all examples, the National Renewable
Energy Laboratory (NREL) 5-MW reference wind turbine (Jonkman et al., 2009) is used. The examples below are provided
on the GitHub repository where the YAMS package is provided (Branlard, 2021).

## 4.2   Notations

We adopt a system of notations where the first letter of a body is used to identify the parameters of that body. As an example,
the tower is represented with the letter T, and the following body parameters are defined: $T$, origin; $M_T$, mass; $L_T$, length;
$(J_{x,T}, J_{y,T}, J_{z,T})$, diagonal coefficients of the inertia tensor about the center of gravity and in body coordinates; $\boldsymbol{r}_{TG}$, vector
from body origin to body center of mass, of coordinates $(x_{TG}, y_{TG}, z_{TG})$ in body coordinates. We also define $\theta_t$, the nacelle
tilt angle about the $y$ axis; $g$, the acceleration of gravity along $-z$; and $O$, the origin of the global coordinate system.

## 4.3   Rotating blade with centrifugal stiffening

We begin with the study of a flexible blade of length $L_B = R$, rotating at the constant rotational speed $\Omega$. We use this test
case to familiarize the reader with the key concepts of the shape function approach given in Appendix B. A sketch of the
system is given in Figure 2. We start by modeling the blade using a single shape function, assumed to be directed along the
$x$-axis ("flapwise"): $\boldsymbol{\Phi}_1 = \Phi\boldsymbol{e}_x$, where $\boldsymbol{e}_x$ is the unit vector in the $x$ direction. The undeflected blade is directed along the radial

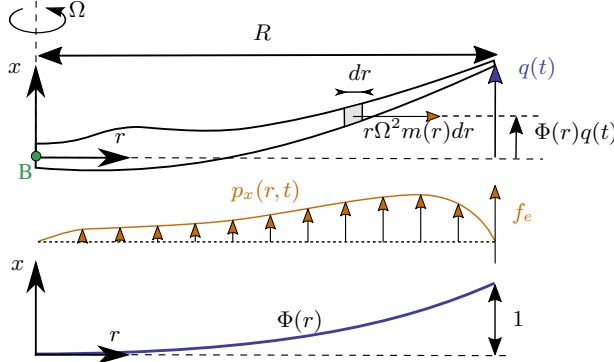

**Figure 2.** Sketch of a rotating blade with the restoring centrifugal force. Points are indicated in green, degrees of freedom in blue, and loads in orange.


coordinate $r$ and rotates around the $x$-axis. We assume that the shape function is known, noted $\Phi(r)$. It can be computed as the
first flapwise mode of the blade using tools provided in YAMS. The expression $\Phi(r) = r^3$ is a simple approximation that can
be used for hand calculations. The aerodynamic force per length in the flapwise direction is noted $p_x(r)$. The generalized mass
and stiffness are computed based on the mass per length ($m$) and flapwise bending stiffness ($EI_y$) of the blade, according to
Equation B1:
$$M_e = \int_0^R m(r)\Phi^2(r)\,dr \tag{26}$$
$$K_e = \int_0^R EI_y(r)\left[\frac{d^2\Phi}{dr^2}(r)\right]^2 dr \tag{27}$$
The generalized force is obtained from Equation B3:
$$f_e = \int_0^R p_x(r,t)\Phi(r)\,dr \tag{28}$$
The important consideration for this model is the axial load, $N$. The main axial load at a radial station $r$ comes from the
centrifugal force acting on all the points outboard of the current station:
$$N(r) = \int_r^R m(r')\Omega^2 r'\,dr' \tag{29}$$
The geometric stiffness contribution of the axial load is obtained from Equation B5 as:
$$K_g(\Omega) = \int_0^R N(r)\left[\frac{d\Phi}{dr}\right]^2 dr = \Omega^2 \int_0^R \int_r^R m(r')r'\,dr' \left[\frac{d\Phi}{dr}\right]^2 dr \tag{30}$$
The geometric stiffness, $K_g$, is positive and increases with the square of the rotational speed. This restoring effect is referred to
as "centrifugal stiffening." In this example, the beam rotates with respect to a fixed support, the influence of gravity is omitted,
and no force other than the centrifugal force is assumed in the radial direction (the Coriolis force contribution to the radial
force is assumed to be negligible for simplicity). Therefore, the only geometric stiffness comes from the centrifugal force.
For a wind turbine blade mounted on a flexible support and under the influence of gravity, the different geometric stiffening
terms presented in Appendix C should be used. Adding the elastic and geometric stiffness, the natural frequency of the blade
increases with the rotational speed as follows:
$$\omega_0(\Omega) = \sqrt{\frac{(K_e + K_g(\Omega))}{M_e}} = \sqrt{\omega_0^2(0) + \frac{K_g(\Omega)}{M_e}} = \sqrt{\omega_0^2(0) + k_\Omega \Omega^2} \tag{31}$$
where $k_\Omega$ is referred to as the "rise factor" or "Southwell coefficient," and in our approximation, it is found to be constant:
$k_\Omega = K_g(\Omega)/M_e/\Omega^2$. The coefficient provides the variation of the blade frequency with rotational speed, which is something
that is observed on a Campbell diagram when performing stability analyses. In general, the mode shapes of the blade will also
change as a function of the rotational speed, and different shape functions should preferably be used for simulations at different
rotational speeds. The effect is fairly limited, and most OpenFAST practitioners only use one shape function corresponding to
the value at rated rotational speed. Similarly, the Southwell coefficient is a function of the rotational speed, but the variation is
negligible as long as the rotational speed is small compared to the natural frequency (e.g., $(\Omega/\omega)^2 \lesssim 5$; see Bielawa (2006)),
which is the case for wind energy applications.
The treatment for a shape function in the edgewise direction is similar, using $\mathbf{\Phi}_2 = \Phi_2 \mathbf{e}_\theta$, where $\mathbf{e}_\theta$ is the unit vector
in the edgewise direction. In this case, the centrifugal force also has a component in the tangential direction, $p_{\theta,\text{centri}}(r) =$
$-\Omega^2 u_\theta(r) dm(r)$, with $u_\theta = \Phi_2 q$. This leads to a generalized force equal to $\int_0^L p_{\theta,\text{centri}} \Phi_2 dr = -\Omega^2 M_e q$, or, equivalently, to a
stiffness term: $K_\omega = -\Omega^2 M_e$. It can be verified that this generalized force corresponds to the contribution $O_{e,11}\omega_x^2$, from $\mathbf{k}_{\omega,e}$,
given in Equation A10. For an edgewise mode, the frequency therefore evolves as:
$$\omega_0(\Omega) = \sqrt{\frac{(K_e + K_g(\Omega) + K_\omega(\Omega))}{M_e}} = \sqrt{\omega_0^2(0) + (k_\Omega - 1)\Omega^2} \tag{32}$$

with $k_\Omega = K_g(\Omega)/M_e/\Omega^2$ and with $K_g$ computed using Equation 30.
We apply the method to the NREL 5-MW wind turbine using the blade properties and shape functions provided in the Elas-
toDyn input file. We order the degrees of freedom as 1st flap, 1st edge, and 2nd flap, assuming no coupling between the shape
functions, so that each can be treated individually using the results from this section. The diagonal coefficients of the mass ma-
trix are $\text{diag}(\mathbf{M}_e) = [9.5e3,\ 1.5e4,\ 5.7e3]$, and for the stiffness matrix they are $\text{diag}(\mathbf{K}_e) = [1.7e4,\ 6.7e4,\ 8.7e4]$, computed
according to Equations 26 and 27. The coefficients $k_\Omega$ of each degree of freedom are obtained as $\mathbf{k}_\Omega = [1.7,\ 1.4,\ 5.5]$. We
compare the frequencies obtained with the present method against OpenFAST linearization results in Figure 3. The simulations
     were run in vacuum (no gravity, no aerodynamics) and with a cone angle of 0 deg. Strong agreement is found for the evolution

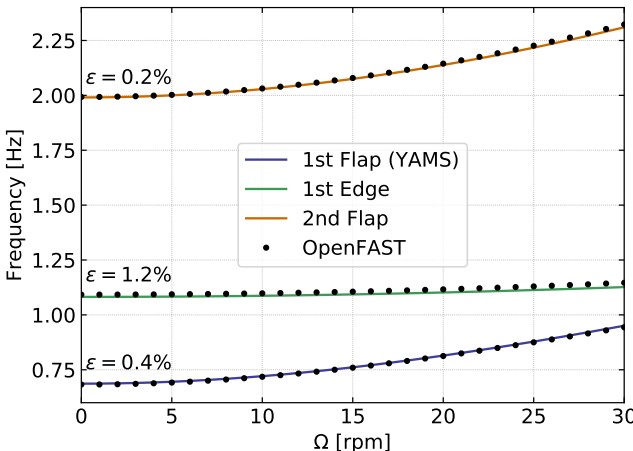

**Figure 3.** Variation of the natural frequencies of the NREL 5-MW turbine blade with rotational speed. Results from YAMS and OpenFAST, with mean relative error, $\epsilon$, are reported on the figure.

of the different frequencies with the rotational speed. The stiffening is less pronounced for edgewise modes as a result of the
softening introduced by $K_\omega$.
This section focused on the analysis of individual shape functions. In the general case, multiple shape functions are present
and couplings might exist between them (due to the structural twist or nonorthogonality of the shape functions, or if the shape
functions have components in multiple directions such as $\boldsymbol{\Phi_1} = \Phi_{1x}\boldsymbol{e}_x + \Phi_{1y}\boldsymbol{e}_y$). In such a case, the general developments of
Appendix A and Appendix B should be used.

### 4.4 Two degrees of freedom model of a land-based or fixed-bottom turbine

We consider a system of three bodies: tower (or support structure), nacelle, and rotor. The system represents a land-based

wind turbine or a fixed-bottom offshore wind turbine. A sketch of the system is given in Figure 4. The nacelle and rotor

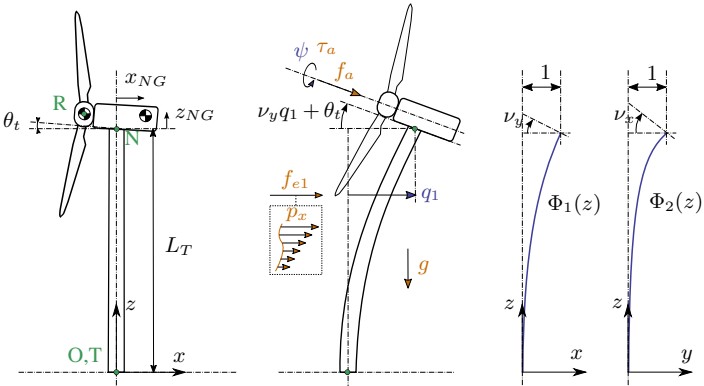

**Figure 4.** Model of a land-based or fixed-bottom wind turbine using one to three degrees of freedom (fore-aft and side-side flexibility of the support structure, and shaft rotation). Points are indicated in green, degrees of freedom in blue, and loads in orange.


blades are rigid bodies, whereas the tower is flexible and represented by one shape function[2] in the fore-aft direction, noted
$\boldsymbol{\Phi_1} = \Phi_1 \boldsymbol{e}_x$. For hand calculations and as a first approximation, the first mode shape of a massless beam with a top mass
may be used: $\Phi_1(z) = 1 - \cos(z\pi/L/2)$. Increased accuracy is obtained when the shape function matches the actual first
tower fore-aft bending mode, accounting for the effect of the rotor-nacelle mass and inertia. The degrees of freedom are
$\boldsymbol{q} = (q, \psi)$, where $q$ is the generalized (elastic) coordinates in the fore-aft direction and $\psi$ is the azimuthal position. The
slope of the tower shape function at the tower top is a key coupling parameter of the model, noted $\nu_y$. When the tower
deflects 1 m in the $x$ direction, the nacelle rotates by an angle $\nu_y$. The method assumes that the tower-top point remains
along the $x$-axis, neglecting the so-called nonlinear geometric effect. However, nonlinear geometric effects can be included
using geometric stiffening corrections (see Appendix C or Branlard (2019)). The aerodynamic thrust and torque are noted
$f_a$ and $\tau_a$, respectively, and act at the rotor center (point $R$). The low-speed shaft generator torque is written as $\tau_g$. The
distributed loads on the tower, $p_x$ (from aerodynamics and hydrodynamics), are projected against the shape function to obtain
the generalized forces $f_e = \int_0^{L_T} p_x(z,t)\Phi_1(z)dz$. The moments of inertia of the rotor in its coordinates are $(J_{x,R}, J_{\oplus,R}, J_{\oplus,R})$.
We note that $M_e, K_e$, and $D_e$ are the generalized mass, stiffness, and damping, respectively, associated with a given shape
function $M_e = \int_0^{L_T} m(z)\Phi_1^2(z)dz$ , $K_e = \int_0^{L_T} EI(z)\left[\frac{d^2\Phi_1}{dz^2}(z)\right]^2 dz$ , $D_e = 2\zeta M_e \omega_e$. where $m(z)$ and $EI(z)$ are the mass
per length and bending stiffness of the tower, respectively, and $\omega_e$ and $\zeta$ are the frequency and damping ratio, respectively,

---

[2]The relevant equations of the shape function approach for a beam are given in Appendix B.

associated with the shape function (assuming the shape function approximates a mode shape). The geometric softening of the tower due to the tower-top mass ($K_{gt}$) and its own weight ($K_{gw}$) is obtained using Equation B5, as $K_g = K_{gt} + K_{gw}$, with :

$$K_{gt} = -g \int_0^{L_T} (M_R + M_N) \left[ \frac{d\Phi_1}{dz}(z) \right]^2 dz \tag{33}$$

$$K_{gw} = -g \int_0^{L_T} \left[ \frac{d\Phi_1}{dz}(z) \right]^2 \left[ \int_z^{L_T} m(z')\, dz' \right] dz \tag{34}$$

The tower is assumed to be fixed and under no significant vertical external loads and therefore the only geometric stiffness comes from the gravitational force. For a tower mounted on a moving support (fixed-bottom foundation or floater), additional geometric stiffening terms would be present (see Appendix C). The shape function frequency is obtained as:

$$\omega_e = \sqrt{(K_e + K_g)/M_e} \tag{35}$$

The application of the symbolic framework leads to the following equations of motion (rearranged for interpretability):

$$\begin{bmatrix} M_q & 0 \\ 0 & J_{x,R} \end{bmatrix} \begin{bmatrix} \ddot{q} \\ \ddot{\psi} \end{bmatrix} = \begin{bmatrix} f_q \\ \tau_a - \tau_g \end{bmatrix} \tag{36}$$

where:

$$M_q = M_e + M_N + M_R \tag{37}$$

$$+ \left( J_{yN} + J_{\oplus,R} + M_N(x_{NG}^2 + z_{NG}^2) + M_R(x_{NR}^2 + z_{NR}^2) \right)\nu_y^2 \tag{38}$$

$$+ 2\left[ (M_N z_{NG} + M_R z_{NR})\cos(\nu_y q) - (M_N x_{NG} + M_R x_{NR})\sin(\nu_y q) \right]\nu_y \tag{39}$$

and

$$f_q = f_e - (K_e + K_g)q - D_e \dot{q} \tag{40}$$

$$+ g\nu_y \left[ (M_N x_{NG} + M_R x_{NR})\cos(\nu_y q) + (M_N z_{NG} + M_R z_{NR})\sin(\nu_y q) \right] \tag{41}$$

$$+ \nu_y^2 \dot{q}^2 \left[ (M_N x_{NG} + M_R x_{NR})\cos(\nu_y q) + (M_N z_{NG} + M_R z_{NR})\sin(\nu_y q) \right] \tag{42}$$

$$+ f_a \nu_y (x_{NR}\sin\theta_t + z_{NR}\cos\theta_t) \tag{43}$$

$$+ f_a \cos(\theta_t + \nu_y q) \tag{44}$$

Details on the derivations are given in Appendix E1. The mass matrix consists of three main contributions: Equation 37 represents the elastic mass and the rotor nacelle assembly (RNA) mass, Equation 38 is the generalized rotational inertia of the RNA, and Equation 39 is the inertial coupling between the tower bending and the rotation of the nacelle. The forcing terms are identified as follows: Equation 40 consists of the elastic load resulting from the external forces on the tower, the elastic and geometric stiffness loads, and the damping load on the tower; Equation 41 is the gravitational load from the RNA, which will

contribute to the stiffness of the system; Equation 42 is the centrifugal force of the RNA (“$M\omega^2 r$” with $\omega = \nu_y \dot{q}$); Equation 43
is the generalized torque from the aerodynamic thrust; and Equation 44 is the thrust contribution acting directly along the
direction of the shape function degree of freedom (along $x$). The RNA center of mass plays an important part in the equations
(see the terms $(M_N x_{NG} + M_R x_{NR})$ and $(M_N z_{NG} + M_R z_{NR})$).
The equations of motion given in Equation 36 can be used to perform time domain simulations of a wind turbine. It is noted
that the two degrees of freedom are only coupled by the aerodynamic loads. The nonlinear model was used in previous work
for time domain simulations and its linear version was used for state estimations (Branlard et al., 2020a, b). In this section,
we apply the linearized form to compute the natural frequency of the turbine tower fore-aft mode. The linearized stiffness is
obtained by taking the gradient of the forcing with respect to $q$, and using a small angle approximation for $\nu_y$ to the second
order:
$$K_{q,lin} = (K_e + K_g) - \nu_y^2 g (M_N z_{NG} + M_R z_{NR} - f_a q \cos\theta_t) + \nu_y f_a \sin\theta_t \tag{45}$$

For the NREL 5-MW reference turbine (Jonkman et al., 2009), the different numerical values are: $g = 9.807 \text{ m}\cdot\text{s}^{-2}$, $\theta_t = 5$
deg, $x_{NR} = -5.0$ m, $z_{NR} = 2.4$ m, $L_T = 87.6$ m, $z_{NG} = 1.75$ m, $x_{NG} = 1.9$ m, $M_R = 1.1e5$ kg, $J_{x,R} = 3.86e7$ kg m$^2$,
$J_{\oplus,R} = 1.92e7$ kg m$^2$, $M_N = 2.4e5$ kg, $J_{y,N} = 1.01e6$ kg m$^2$, $M_{RNA} = 3.5e5$ kg. The first fore-aft shape function of the
NREL 5-MW turbine tower and its derivatives are:
$$\Phi_1(z) = (a_2 \bar{z}^2 + a_3 \bar{z}^3 + a_4 \bar{z}^4 + a_5 \bar{z}^5 + a_6 \bar{z}^6)/(a_2 + a_3 + a_4 + a_5 + a_6)$$

$$\frac{d\Phi_1}{dz}(z) = \frac{1}{L_T}(2a_2 \bar{z} + 3a_3 \bar{z}^2 + 4a_4 \bar{z}^3 + 5a_5 \bar{z}^4 + 6a_6 \bar{z}^5)/(a_2 + a_3 + a_4 + a_5 + a_6) \tag{46}$$

$$\frac{d^2\Phi_1}{dz^2}(z) = \frac{1}{L_T^2}(2a_2 + 6a_3 \bar{z} + 12a_4 \bar{z}^2 + 20a_5 \bar{z}^3 + 30a_6 \bar{z}^4)/(a_2 + a_3 + a_4 + a_5 + a_6)$$

with $\bar{z} = z/L$, $a_2 = 0.7004$, $a_3 = 2.1963$, $a_4 = -5.6202$, $a_5 = 6.2275$, and $a_6 = -2.504$. The material properties and the shape
function are illustrated in Figure 5. The scaling of the shape functions given in Equation 46 is important to obtain the correct
numerical values for the flexible tower, namely: $\nu_y = 0.0185$, $M_e = 5.4e4$, $K_e = 1.91e6$, $K_g = -5.2e4 - 1.0e4 = -6.20e4$,
$\omega_e = \sqrt{(K_e + K_g)/M_e} = 5.85$ rad/s. These numerical values, with $q = 0$, lead to: $M_q = 4.375e5$ and $K_q = 1.849e9$. The
first fore-aft mode of the wind turbine has a natural frequency of $f = \sqrt{K_q/M_q} = 0.3272$ Hz. This value was compared with
results obtained using OpenFAST linearization. Both methods are in strong agreement, with differences only arising at the fifth
decimal place.
**4.5 Three-degrees-of-freedom model of a land-based or fixed-bottom turbine**
We consider the same system as the one presented in subsection 4.4, but the tower is now represented by one shape function in
both the fore-aft and side-side directions, $\mathbf{\Phi}_1 = \Phi_1 \mathbf{e}_x$ and $\mathbf{\Phi}_2 = \Phi_2 \mathbf{e}_y$. The degrees of freedom are $\mathbf{q} = (q_1, q_2, \psi)$, where $q_1$
and $q_2$ are the generalized (elastic) coordinates in the fore-aft and side-side directions, respectively, and $\psi$ is the rotor azimuth.
A sketch of the system is given in Figure 4.

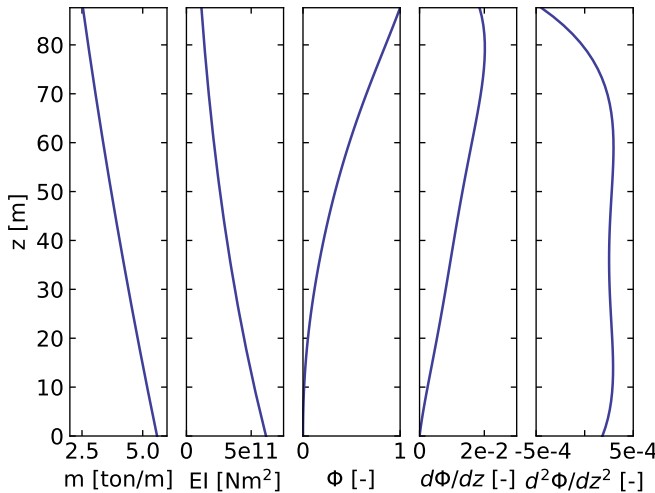

**Figure 5.** Properties of the NREL 5-MW turbine tower: mass per length ($m$), bending stiffness ($EI$), and shape function displacement ($\Phi$), slope ($d\Phi/dz$) and curvature ($d^2\Phi/dz^2$).

The slopes of the shape functions at the tower top are key coupling parameters of the model, noted $\nu_x$ and $\nu_y$. The aerody-
namic thrust and torque are noted $f_a$ and $\tau_a$, acting at point $R$. The distributed loads on the tower, $p_x$ and $p_y$ (from aerodynam-
ics and hydrodynamics), are projected against the shape functions to obtain the generalized forces $f_{e1} = \int \Phi_1 p_x dz$ and $f_{e2} =$
$\int \Phi_2 p_y dz$. The moments of inertia of the rotor in its coordinates are $(J_{x,R}, J_{\oplus,R}, J_{\oplus,R})$. We note that $\boldsymbol{M}_e$, $\boldsymbol{K}_e$, and $\boldsymbol{D}_e$ are
the generalized mass, stiffness, and damping, respectively, associated with a given shape function (e.g., $M_{e11} = \int \Phi_1^2 m(z) dz$,
where $m$ is the mass per length of the tower). The application of the symbolic framework leads to the equations of motion
given in Appendix E2. To simplify the equations and limit their length when printing them in this article, we have applied a
first-order small-angle approximation for $\theta_t$, and a second-order approximation for $\nu_x$ and $\nu_y$. It is observed from Equation E14
that a first-order approximation for $\nu_y$ would have removed the influence of the rotor and nacelle $y$-inertia on the generalized
mass associated with the tower fore-aft bending.
We performed a time simulation of the model using both our symbolic framework YAMS and OpenFAST. The time integra-
tion in YAMS currently relies on tools provided in the SciPy package, which implements several time integrators. A sufficient
level of accuracy was obtained using a fourth-order Runge-Kutta method, which is the default method. Kane's method, which
uses a minimal set of coordinates, tends to lead to stiff systems, and it is possible that implicit integrators may be needed for
other systems. We compare the time series obtained using our generated functions with results from the equivalent OpenFAST
simulation in Figure 6. In this simulation, the tower top is initially displaced by $1\,\mathrm{m}$ in the $x$ and $y$ directions, and the rotational
speed is $5\,\mathrm{rpm}$. We report the mean relative error, $\epsilon$, and the coefficient of determination, $R^2$, on the figure. We observe that
our model is in strong agreement with the OpenFAST simulation. The differences in the second tower degree of freedom are
attributed to 1) the handling of the small-angle approximation, which is different in OpenFAST (using the closest orthonormal
matrix; Jonkman (2009)) and in our formulation (two successive rotations, linearized); 2) the nonlinear geometric corrections

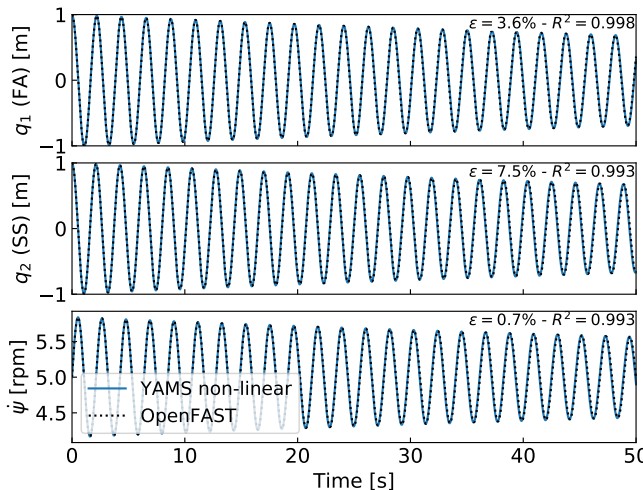

**Figure 6.** Free decay results for the land-based/fixed-bottom model using both the symbolic framework (YAMS) and OpenFAST. From top to bottom: tower fore-aft bending, tower side-side bending, and shaft rotational speed.

that are implemented in OpenFAST, which we have omitted here by only selecting shape function expansion to the zeroth order (see subsection 5.2). The variation in azimuthal speed, resulting from the coupling between the gyroscopic loads and the tower bending, is captured well.

### 4.6  Three-degrees-of-freedom model of a floating wind turbine

In this example, we demonstrate the applicability of the method for a floating wind turbine. We model the turbine using three bodies: rigid floater, flexible tower, and rigid RNA (labeled "N"). The degrees of freedom selected are: $q = (x, \phi, q_T)$, where $x$ is the floater surge, $\phi$ is the floater pitch, and $q_T$ is the coordinate associated with a selected fore-aft shape function. A sketch of the model is given in Figure 7. The notations are similar to the ones presented in subsection 4.5. Lumped hydrodynamic loads at the floater center of mass are now added. The model can also be used for a combined tower and floater that is flexible, simply by setting the mass of the floater to zero and including the hydrodynamic loading into the loading $p_x$. The equations of motion are given in Appendix E3. The equations were simplified using a first-order small-angle approximation of $\theta_t$ and $\phi_y$, and a second-order approximation for $\nu_y$.

We performed a numerical simulation of the model generated by YAMS and compared it with OpenFAST for a case with gravitational loads only, starting with $x = 0$ m, $\phi = 2$ deg, and $q_T = 1$ m. The results are presented in Figure 8. We observe again that the results from the two models correlate to a high degree.

We also compared the linearized version of both models. The symbolic framework can generate the linearized mass, stiffness, and damping matrices, as described in subsection 2.5. The matrices are then combined into a state matrix and compared with the state matrices written by the OpenFAST linearization feature. The eigenvalue analysis of the YAMS state matrix returned

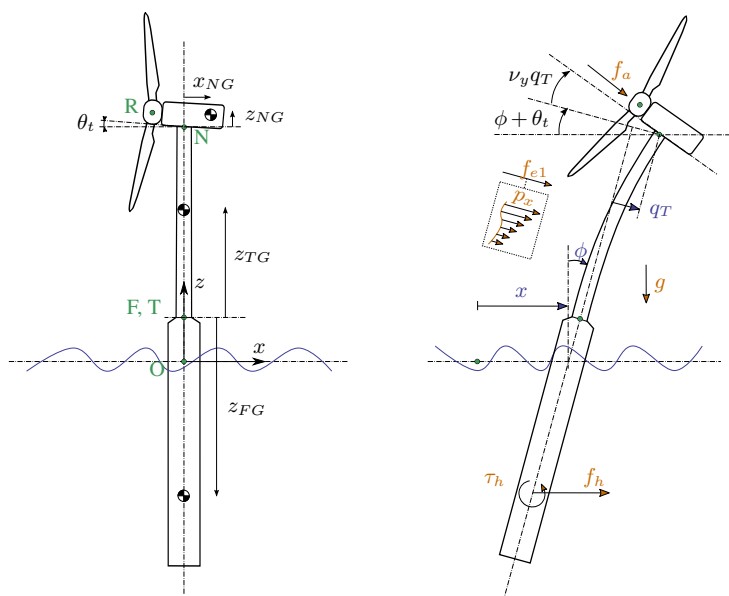

**Figure 7.** Model of a floating wind turbine using three degrees of freedom. Points are indicated in green, degrees of freedom in blue, and loads in orange.

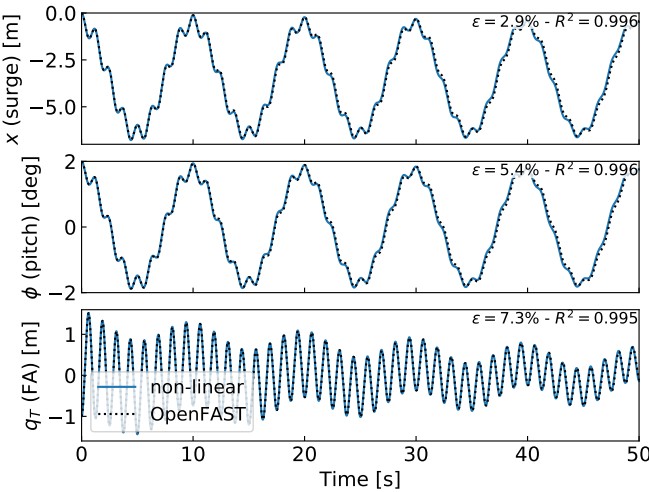

**Figure 8.** Free-decay results for the floating wind turbine model using YAMS and OpenFAST. From top to bottom: surge, pitch, and tower fore-aft bending.

a pitch and fore-aft frequencies of $0.099$ Hz and $0.799$ Hz, respectively, whereas OpenFAST returned $0.095$ Hz and $0.795$ Hz.
The $4\%$ error in the pitch frequency appears reasonable in view of the approximations used.

# 5 Discussions

## 5.1 Applications and advantages of the method

The implementation of the symbolic YAMS library was originally motivated by the need to obtain a simple linearized model of a floating wind turbine for frequency domain simulations. There are multiple potential applications of the framework:

- The generated equations can be used in time domain simulation tools. The equations can be readily exported to different programming languages (C, FORTRAN, or Python) providing computationally efficient tools, particularly because the method generates compact and minimal equations. This is in contrast to most other multibody codes, in which many terms are calculated as matrix equations and through successive function calls. Further, the symbolic framework allows us to generate optimized code, in which common terms and factors are computed once and stored in temporary variables for reuse in the different expressions. In our examples, time domain simulations were observed to be 2 orders of magnitude faster when using the automatically generated code in Python compared to OpenFAST simulations that rely on a compiled language. Using such a framework can be considered in the future to replace the existing ElastoDyn module of OpenFAST. It can also be applied to unusual configurations such as multirotor or vertical-axis turbine concepts. Dedicated code can be generated for specific applications for increased performance. For instance, implicit integrators with iterative Newton-Raphson-like solvers benefit from the possibility of generating exact and efficient Jacobians along with the equations of motion.

- The generation of linearized models has a wide range of applications, such as linear time domain simulations, controller design and tuning, frequency domain analyses, stability analyses, state observers, or digital twins. The symbolic approach is severalfold faster than alternative approaches because it can be evaluated for all operating points at once, whereas other methods (e.g., OpenFAST, HAWCStab2) require multiple linearization calls.

- Analytical linearization with respect to parameters is directly obtained using our tool, which can be used for sensitivity analyses, parameter studies, optimizations, integrated design approaches, and controls co-design (e.g., using methods such as linear-matrix-inequality-based designs) (Pöschke et al., 2020). Nonanalytical approaches require numerous linearizations and evaluations at various operating points (Jonkman et al., 2022).

- In addition to the nonlinear or linear equations of motion in minimal coordinates, the equations for the constraint forces or any auxiliary kinematic variable can also be generated efficiently by inserting unknown virtual displacements in the equations (see Appendix D5 for an alternative approach). The position of all bodies in local or global coordinates can be recovered from the minimal coordinates and, in combination with the flexible code generation, be used to output data (e.g., for 3D animations of the turbine).

- Analytical gradients of the equations can be computed and used in optimizations, nonlinear model predictive control, or moving horizon estimation. External loads that cannot be expressed analytically can be defined as generic functions

of the structural degrees of freedom, inputs, and parameters. After the code generation, the user can link a numerical implementation of the function and its numerical gradients to be able to use a mix of analytical and numerical gradients.

– Another advantage of the presented method is the possibility to quickly generate models with different levels of detail, ensuring consistency between the different levels of fidelity. This is in contrast to other more heuristic modeling approaches in which parameters often have to be retuned for each added degree of freedom.

– The method provides useful insights and can be used as an educational tool: simple models of a system with few degrees of freedom can readily be obtained, studied, and compared to hand-based calculation.

## 5.2 Advanced consideration

Section 2 addressed the systematic derivation of the equations of motion for an assembly of rigid or flexible bodies. Some advanced aspects of the method are discussed here:

– The different terms involved in the equations of motion of flexible bodies can be decomposed using shape integrals (see Appendix D3). Our framework readily supports this optional decomposition: it is the responsibility of the user to provide the terms and values of the expansion when numerical evaluation is to occur.

– The definition of geometric stiffening requires attention in the general case. It is accounted for by the term $k_\sigma$, presented in Appendix A. We discuss geometric stiffening in more detail in Appendix C.

– The treatment of external loads was not addressed in detail in this article because the loads are application-specific (aerodynamics, hydrodynamics, etc.). The framework can accept external loads as arbitrary functions of multiple variables or as analytical expressions. In the former case, the user will have to provide an implementation of the function during the execution.

– Even though the equations of motion are void of constraint forces, the values of these forces can be recovered. They can be expressed as functions of the external forces and the states of the system. It is not necessary to compute them by iteratively solving constraint equations.

– The framework can easily include rheonomous constraints—for instance, for the pitch angle—without having to supply a dedicated torque. Pitch speed and accelerations can be directly introduced into the mechanical system if they are provided by a generic second-order pitch actuator model.

## 5.3 Limitations

In spite of the advantages listed in subsection 5.1, the symbolic procedure presented in this work has some potential limitations:

– Constraints and closed loops have currently not been added to the framework. The SymPy mechanics package supports additional constraint equations within Kane's method. We therefore hope that this limitation can be lifted in the future.

– Large problems may challenge a symbolic calculation package: memory impact, calculation time, simplification times,
and size of expressions may become significant. Some of these issues may be alleviated by introducing intermediate
variables that are only substituted for in the numerical implementation or by using a recursive formulation of the solution
procedure (Branlard, 2019).

We further note that the shape function approach is an approximate method: it introduces a separation of space and time early
on in the development of the nonlinear equations of motion, and applies low order polynomial (usually linear or quadratic)
approximations to eliminate high-order terms (see e.g. Table 1 of Wallrapp (1994)). This was presented as an advantage
in subsection 5.1 because the equations are obtained in compact form and are readily linearized. Yet, the approximations
introduced by the method may imply that nonlinearities are not well captured, which is why the models are labeled as "mid-
fidelity" throughout this article. The domain of validity of the nonlinear or linear models presented may therefore be limited in
time and space as opposed to fully nonlinear methods. Advanced methods to obtain high-fidelity reduced-order models from
nonlinear dynamic systems are beyond the scope of this work, see, e.g. Steindl and Troger (2001); Benner et al. (2015); Touzé
et al. (2021).

## 469   6   Conclusions

We presented a symbolic framework to obtain the linear and nonlinear equations of motion of a multibody system made of
rigid bodies, flexible bodies, and kinematic joints. Our approach is based on Kane's method and a nonlinear shape function
representation of flexible bodies. We provided different wind energy examples and verified the results against OpenFAST
simulations. The framework can readily provide models suitable to a wide range of applications with competitive computational
times. The framework is open source, and the examples presented are available in the repository. Future work will focus on
applying the framework to dedicated research projects, with more complex systems, and potentially extend the framework to
account for closed-loop systems and arbitrary constraints.
*Author contributions.*   Both authors exchanged over the last two years about the implementation of such a framework and its application to
wind energy. EB wrote a Python implementation and JG wrote a Maxima implementation. EB wrote the main corpus of the article, with
feedback and contributions from JG.
*Competing interests.*   No competing interests are present.
*Code availability.*   A Zenodo link will be created for https://github.com/ebranlard/yams. The examples given in this articles are found in the
folder `welib/yams/papers` of the repository.
*Acknowledgements.* This work was authored in part by the National Renewable Energy Laboratory, operated by Alliance for Sustainable
Energy, LLC, for the U.S. Department of Energy (DOE) under Contract No. DE-AC36-08GO28308. Funding provided by U.S. Department
of Energy Office of Energy Efficiency and Renewable Energy Wind Energy Technologies Office. The views expressed in the article do
not necessarily represent the views of the DOE or the U.S. Government. The U.S. Government retains and the publisher, by accepting the
article for publication, acknowledges that the U.S. Government retains a nonexclusive, paid-up, irrevocable, worldwide license to publish or
reproduce the published form of this work, or allow others to do so, for U.S. Government purposes.
*Financial support.* This work was funded under the Technology Commercialization Fund Project, supported by the DOE's Wind Energy
Technologies Office.

## 491 Appendix A: Equations for a flexible body and shape integrals

In this section, we detail the equations of motion of a flexible body. The reader is referred to the following references for a complete treatment
of the equations of motion: Shabana (2013), Schwertassek and Wallrapp (1999), and Wallrapp (1994). The subscript $i$, indicating the body
index, is dropped. All quantities (vectors and matrices) are expressed in the body frame of reference; therefore, the prime notation is also
dropped in this section. The number of flexible shape functions associated with the body is $n_e$, the flexible degrees of freedom are $q_e$, and
the shape functions are gathered into a matrix $\mathbf{\Phi}$ of size $(3 \times n_e)$. The equations of motion, given in Equation 11, are repeated below:

$$
\qquad
\begin{bmatrix} \boldsymbol{M}_{xx} & \boldsymbol{M}_{x\theta} & \boldsymbol{M}_{xe} \\ & \boldsymbol{M}_{\theta\theta} & \boldsymbol{M}_{\theta e} \\ \text{sym.} & & \boldsymbol{M}_{ee} \end{bmatrix}
\begin{bmatrix} \boldsymbol{a}_i \\ \dot{\boldsymbol{\omega}}i \\ \ddot{\boldsymbol{q}}_e \end{bmatrix}
+
\begin{bmatrix} \boldsymbol{k}_{\omega,x} \\ \boldsymbol{k}_{\omega,\theta} \\ \boldsymbol{k}_{\omega,e} \end{bmatrix}
+
\begin{bmatrix} 0 \\ 0 \\ \boldsymbol{k}_e \end{bmatrix}
=
\begin{bmatrix} \boldsymbol{f}_x \\ \boldsymbol{f}_\theta \\ \boldsymbol{f}_e \end{bmatrix}
\qquad \text{(A1)}
$$

The different terms of the mass matrix are obtained as follows:

$$
\quad \boldsymbol{M}_{xx} = \int \boldsymbol{I}_3 \, \mathrm{d}m = M\boldsymbol{I}_3 \qquad (3 \times 3) \tag{A2}
$$

$$
\quad \boldsymbol{M}_{x\theta} = - \int \tilde{\boldsymbol{s}}_P \, \mathrm{d}m = -M\tilde{\boldsymbol{s}}_{CM} \qquad (3 \times 3) \tag{A3}
$$

$$
\quad \boldsymbol{M}_{\theta\theta} = - \int \tilde{\boldsymbol{s}}_P \tilde{\boldsymbol{s}}_P \, \mathrm{d}m = \boldsymbol{J} \qquad (3 \times 3) \tag{A4}
$$

$$
\quad \boldsymbol{M}_{\theta e} = \int \tilde{\boldsymbol{s}}_P \boldsymbol{\Phi} \, \mathrm{d}m = \boldsymbol{C}_r^T \qquad (3 \times n_e) \tag{A5}
$$

$$
\quad \boldsymbol{M}_{xe} = \int \boldsymbol{\Phi} \, \mathrm{d}m = \boldsymbol{C}_t^T \qquad (3 \times n_e) \tag{A6}
$$

$$
\quad \boldsymbol{M}_{ee} = \int \boldsymbol{\Phi}^T \boldsymbol{\Phi} \, \mathrm{d}m \qquad (n_e \times n_e) \tag{A7}
$$

The integrals are volume integrals over the volume of the body (for beams, they reduce to line integrals). The notation $[\tilde{\ }]$ represents the skew
symmetric matrix. $M$ is the mass of the body. The vector $\boldsymbol{s}_{CM}$ is the vector from the origin of the body to undeflected center or mass (CM)
of the body. The notations $\boldsymbol{C}_t$ $(n_e \times 3)$ and $\boldsymbol{C}_r$ $(n_e \times 3)$ are introduced to match Wallrapp's notations. The vector $\boldsymbol{s}_P$ is the vector from
the origin of the body to a deflected point of the body of elementary mass $\mathrm{d}m$. The undeflected position of this point is written as $\boldsymbol{s}_{P_0}$ and
the displacement field $\boldsymbol{u}$, such that: $\boldsymbol{s}_P = \boldsymbol{s}_{P_0} + \boldsymbol{u}$. Typically, the displacement field is given by $\boldsymbol{u} = \boldsymbol{\Phi}\boldsymbol{q}_e$, but a higher-order expansion can
also be introduced (see Wallrapp (1994) and Appendix D4). Wallrapp also includes the elementary mass moment of inertia, which results in
additional terms in the integrals (see Wallrapp (1994)). Such contributions are relevant, for instance, when considering the torsion of a beam
(see Branlard (2019)). The block matrices $\boldsymbol{M}_{xx}$, $\boldsymbol{M}_{xe}$, and $\boldsymbol{M}_{ee}$ do not depend on the deformation of the body and are therefore constant.
The other terms are functions of $\boldsymbol{q}_e$. They may be expressed as linear combinations of constant integrals (see Appendix D3).
The quadratic velocity terms, $\boldsymbol{k}_\omega$, are given as:
$$\boldsymbol{k}_{\omega,x} = 2\tilde{\boldsymbol{\omega}}\boldsymbol{C}_t^T\dot{\boldsymbol{q}}_e + M\tilde{\boldsymbol{\omega}}\tilde{\boldsymbol{\omega}}\boldsymbol{s}_{CM} \qquad (3 \times 1) \tag{A8}$$
$$\boldsymbol{k}_{\omega,\theta} = \tilde{\boldsymbol{\omega}}\boldsymbol{M}_{\theta\theta}\boldsymbol{\omega} + \left[\sum_{j=1..n_e}\boldsymbol{G}_{r,j}\dot{q}_{e,j}\right]\boldsymbol{\omega} \qquad (3 \times 1) \tag{A9}$$
$$\boldsymbol{k}_{\omega,e} = \left[\boldsymbol{\omega}^T\boldsymbol{O}_{e,j}\boldsymbol{\omega}\right]_{j=1..n_e} + \left[\sum_{j=1..n_e}\boldsymbol{G}_{e,j}\dot{q}_{e,j}\right]\boldsymbol{\omega} \qquad (n_e \times 1) \tag{A10}$$
where
$$\boldsymbol{G}_{r,j} = -2\int \tilde{\boldsymbol{s}}_P\tilde{\boldsymbol{\Phi}}_j\,\mathrm{d}m \qquad (3 \times 3) \tag{A11}$$
$$\boldsymbol{O}_{e,j} = \int \tilde{\boldsymbol{\Phi}}_j\tilde{\boldsymbol{s}}_P\,\mathrm{d}m = -\frac{1}{2}\boldsymbol{G}_{r,j}^T \qquad (3 \times 3) \tag{A12}$$
$$\boldsymbol{G}_{e,j} = -2\int \boldsymbol{\Phi}^T\tilde{\boldsymbol{\Phi}}_j\,\mathrm{d}m \qquad (n_e \times 3) \tag{A13}$$
The first term of Equation A10 is obtained by vertically stacking the contribution of each shape function. In the standard input data format,
this term is reshaped as the product $\boldsymbol{O}_e\boldsymbol{\Omega}$, where:
$$\boldsymbol{O}_e = [\boldsymbol{O}_{e,j,11},\ \boldsymbol{O}_{e,j,22},\ \boldsymbol{O}_{e,j,33},\ \boldsymbol{O}_{e,j,12} + \boldsymbol{O}_{e,j,21},\ \boldsymbol{O}_{e,j,23} + \boldsymbol{O}_{e,j,32},\ \boldsymbol{O}_{e,j,13} + \boldsymbol{O}_{e,j,31}]_{j=1..n_e} \qquad (n_e \times 6) \tag{A14}$$
$$\boldsymbol{\Omega} = \left[\omega_x^2,\ \omega_y^2,\ \omega_z^2,\ \omega_x\omega_y,\ \omega_y\omega_z,\ \omega_x\omega_z\right] \qquad (6 \times 1) \tag{A15}$$
The body elastic forces are given by:
$$\boldsymbol{k}_e = \boldsymbol{k}_\sigma + \boldsymbol{K}_e\boldsymbol{q}_e + \boldsymbol{D}_e\dot{\boldsymbol{q}}_e \tag{A16}$$
where $\boldsymbol{K}_e$ and $\boldsymbol{D}_e$ are the elastic stiffness and damping matrices, and $\boldsymbol{k}_\sigma$ represents geometric stiffening terms (see Appendix C). The elastic
damping forces are often given as stiffness proportional damping. For more details, see Wallrapp (1994), and for more examples with elastic
beams, see Branlard (2019). The external loads can be assumed to consist of distributed volume forces, $\boldsymbol{p}$ (in practice they are primarily
surface forces or line forces), and a gravitational acceleration field, $\boldsymbol{g}$. The components of the external loads in Equation A1 are then obtained
by integration over the whole body:
$$\boldsymbol{f}_x = \int \boldsymbol{p}\,dV + \boldsymbol{M}_{xx}\boldsymbol{g} \qquad (3 \times 1) \tag{A17}$$
$$\boldsymbol{f}_\theta = \int \boldsymbol{s}_P \times \boldsymbol{p}\,dV + \boldsymbol{M}_{\theta x}\boldsymbol{g} \qquad (3 \times 1) \tag{A18}$$
$$\boldsymbol{f}_e = \int \boldsymbol{\Phi}^T\boldsymbol{p}\,dV + \boldsymbol{M}_{ex}\boldsymbol{g} \qquad (n_e \times 1) \tag{A19}$$
**Appendix B: Application of the shape function approach to an isolated beam**
In this section, we illustrate how the elastic equations of Appendix A can be applied to an isolated beam. Examples of applications are further
given in subsection 4.3 and subsection 4.4. We consider a beam directed along the $z$-axis and bending in the $x$ and $y$ directions. Expressions
are written in the coordinate system of the beam and primes are dropped in this section. The beam properties are the following: length, $L$,
mass per length, $m$, and bending stiffness, $EI_x$ and $EI_y$. We assume that the displacement field is such that the shape functions are functions
of $z$ only: $\boldsymbol{u}(z,t) = \sum_{i=1}^{n_e} \boldsymbol{\Phi}_i(z) q_{e,i}(t)$. We also assume that the shape functions satisfy at least the geometric boundary conditions. The
kinetic energy of the beam is $T = \frac{1}{2} \int_0^L m \dot{u}^2 dz = \frac{1}{2} \sum_i \sum_j M_{e,ij} \dot{q}_{e,j} \dot{q}_{e,i}$. where $M_{e,ij}$ is (see Equation A7):
$$M_{e,ij} = \int_0^L m(z) \boldsymbol{\Phi}_i(z) \cdot \boldsymbol{\Phi}_j(z)\, dz, \quad i,j = 1, \dots n_e \tag{B1}$$
Equation B1 involves a scalar product of the shape functions at each spanwise position. Integrals over the moment of inertia can be used
to account for torsion (see Branlard (2019)). The potential energy (strain energy) of the beam, is obtained as $V = \frac{1}{2} \sum_i \sum_j K_{e,ij} q_{e,i} q_{e,j}$,
where $K_{e,ij}$ are the elements of the stiffness matrix, which, under the assumption of small deformations, are given by:
$$K_{e,ij} = \int_0^L \left[ EI_y \frac{d^2\Phi_{i,x}}{dz^2} \frac{d^2\Phi_{j,x}}{dz^2} + EI_x \frac{d^2\Phi_{i,y}}{dz^2} \frac{d^2\Phi_{j,y}}{dz^2} \right] dz, \quad i,j = 1, \dots n_e \tag{B2}$$
Elongation and torsional strains ($EA$ and $GK_t$) can similarly be added to the strain energy and the stiffness matrix if longitudinal and
torsional displacement fields are included in the shape functions. The external loads on the beam are assumed to consist of a distributed force
vector, $\boldsymbol{p}(z)$. The virtual work done by the force $\boldsymbol{p}$ for each virtual displacement $\delta q_{e,i}$ provides the generalized force as (see Equation A17):
$$f_{e,i} = \int_0^L \boldsymbol{\Phi}_i \cdot \boldsymbol{p}\, dz \tag{B3}$$
The equations of motion of the isolated beam and then written in matrix form as:
$$\boldsymbol{M}_e \ddot{\boldsymbol{q}}_e + \boldsymbol{D}_e \dot{\boldsymbol{q}}_e + \boldsymbol{K}_e \boldsymbol{q}_e = \boldsymbol{f}_e \tag{B4}$$
where $\boldsymbol{q}_e = [q_{e,1}, \cdots, q_{e,n}]$. Damping is typically added a posteriori to the equations, where the Rayleigh damping assumption is often used:
$\boldsymbol{D}_e = \alpha \boldsymbol{M}_e + \beta \boldsymbol{K}_e$ (stiffness proportional damping implies $\alpha = 0$). If the shape functions are mode shapes, then the shape functions are
orthogonal, the mass and stiffness matrices are diagonal, and the stiffness values would be $K_{e,ii} = \omega_{e,i}^2 M_{e,ii}$, with $\omega_{e,i} = \sqrt{K_{e,ii}/M_{e,ii}}$
the eigenfrequency of the beam mode $i$. The modal damping is then given by $D_{e,ii} = 2\zeta_i M_{e,ii} \omega_{e,i}$, where $\zeta_i$ is the damping ratio associated
with mode $i$.
If the beam is loaded axially by a force $N(z)$ (assumed to be independent of the elastic degrees of freedom), then this force produces a
distributed load in the transverse direction equal to $\boldsymbol{n} = \frac{\partial}{\partial z}\left[ N(z) \frac{\partial \boldsymbol{u}}{\partial z} \right]$, with components in the $y$ and $z$ directions (see Branlard (2019)). The
generalized force associated with this loading is then $Q_{N,i} = \int_0^L \boldsymbol{\Phi}_i \cdot \boldsymbol{n}\, dz$. Inserting the expression of $\boldsymbol{n}$ and $\boldsymbol{u}$, the generalized force has
the form of a stiffness term: $Q_{N,i} = -\sum_j K_{N,ij} q_{e,j}$ with
$$K_{N,ij} = -\int_0^L \boldsymbol{\Phi}_i \cdot \frac{d}{dz}\left[ N(z) \frac{d\boldsymbol{\Phi}_j}{dz} \right] dz = \int_0^L N(z) \frac{d\boldsymbol{\Phi}_i}{dz} \cdot \frac{d\boldsymbol{\Phi}_j}{dz} - \left[ N(z)\boldsymbol{\Phi}_i \cdot \frac{d\boldsymbol{\Phi}_j}{dz} \right]_0^L \tag{B5}$$
and where integration by parts was used to obtain the second equality. Examples of applications are given in subsection 4.3 and subsection 4.4.
The fact that an axial load leads to a stiffness term is referred to as "geometric stiffness," which is the topic of Appendix C.

## Appendix C: Geometric stiffness

### C1  General treatment

Geometric stiffness refers to the apparent change of stiffness of a structure depending on the loading it is subject to. In this section, we present a linear formulation of geometric stiffness for a flexible body undergoing motion and subject to arbitrary loading, inspired by Schwertassek and Wallrapp (1999). Additional details may be found in Wallrapp and Schwertassek (1991). The main component of the geometric stiffening term $\boldsymbol{k}_\sigma$ can be written:

$$\boldsymbol{k}_\sigma = \boldsymbol{K}_g \boldsymbol{q}_e \tag{C1}$$

where $\boldsymbol{K}_g$ is the geometric stiffness matrix of shape $n_e \times n_e$. In general, this matrix is time-dependent, as it is a function of the inertial and external loads acting on the body. The inertial loads consist of contributions from the linear acceleration, $\boldsymbol{a}$, rotational acceleration, $\dot{\boldsymbol{\omega}}$, and cross products of the rotational velocity of the body (centrifugal and gyroscopic terms). The external loads consist of the gravitational force, distributed forces per unit length, $\boldsymbol{p}$, point loads, $\boldsymbol{F}^k$, and point moments, $\boldsymbol{\tau}^k$, where $k$ is the node index where the point loads are applied. Each of these contributions can be computed at each time step using a linear superposition of unit geometric stiffness matrices, noted $\boldsymbol{K}_{g*}$, as follows:

$$\boldsymbol{K}_g = \sum_{\alpha=1}^{3} \left[ (a_\alpha - g_\alpha) \boldsymbol{K}_{gt,\alpha} + \dot{\omega}_\alpha \boldsymbol{K}_{gr,\alpha} \right] + \sum_{\alpha=1}^{3}\sum_{\beta=1}^{3} \omega_\alpha \omega_\beta \boldsymbol{K}_{g\omega,\alpha\beta}$$

$$+ \sum_{\alpha=1}^{3} \left[ p_\alpha \boldsymbol{K}_{gp,\alpha} + \sum_k \left( F_\alpha^k \boldsymbol{K}_{gF,\alpha}^k + \tau_\alpha^k \boldsymbol{K}_{g\tau,\alpha}^k \right) \right] \tag{C2}$$

where the indices $\alpha$ and $\beta$ run on the $x, y$, and $z$ coordinates of the body reference frame. The matrices $\boldsymbol{K}_{g*,\alpha}$ or $\boldsymbol{K}_{g*,\alpha\beta}$ have the shape $n_e \times n_e$ and are obtained as the geometric stiffness matrices for unit accelerations, loads, or products of rotational velocities in the given direction defined by $\alpha$ and $\beta$ ($x$, $y$, or $z$). For instance, $\boldsymbol{K}_{gt,z}$ is the geometric stiffness matrix corresponding to a unit acceleration in the $z$ direction, $\boldsymbol{K}_{g\omega,xy}^k$ is the geometric stiffness matrix corresponding to a unit gyration about the $x$ and $y$ directions (centrifugal effect), and $\boldsymbol{K}_{gF,x}^k$ is the geometric stiffness matrix corresponding to a unit force in the $x$ direction applied at the node $k$ along the body. The effect of the Coriolis force is not mentioned in the work of Schwertassek and Wallrapp and not explicitly accounted for in Equation C2. The Coriolis force, $2m(z)\boldsymbol{\omega} \times (\sum_j \boldsymbol{\Phi}_j(z)\dot{q}_{e,j})$, is proportional to $\dot{\boldsymbol{q}}_e$. Because the instantaneous beam slope is proportional to $\boldsymbol{q}_e$, the geometric stiffening term consists of terms of the form $q_{e,j}\dot{q}_{e,k}$. In Table 1 of Wallrapp (1994), it is stated that nonlinear terms of the form $q_{e,j}\dot{q}_{e,k}$ are neglected. Yet, if the steady state deflection $\boldsymbol{q}_e$ is significant, then the influence of the Coriolis term on the geometric stiffening may be significant. The effect can be included as an additional term in Equation C1 that is a function of $\boldsymbol{q}_e$ and $\dot{\boldsymbol{q}}_e$ (expressions are provided in subsection C2). We note that the terms $\boldsymbol{K}_{g*}$ have different units; for instance, the terms $\boldsymbol{K}_{gt,*}$ are expressed in $\mathrm{N} \cdot \mathrm{s}^2 \cdot \mathrm{m}^{-2}$ .

### C2  Expressions for a beam directed along $z$

The expression for each of these matrices are given in Schwertassek and Wallrapp (1999) in the context of the finite-element method. The general expressions for a shape function approach would be beyond the scope of this article, but we provide the expressions for a beam below.

We adopt the same notations as Appendix B to describe the flexible beam. Following the developments that led to Equation B5, the
geometric correction associated with an axial load $N$, is given by the generalized force:
$$k_{\sigma,N,i} = \int_0^L N(z)\mathbf{\Phi}_i \cdot \left[\sum_j \frac{d\mathbf{\Phi}_j}{dz}q_j\right]dz \tag{C3}$$
When the axial load is not a function of the degrees of freedom, this expression can be expressed as a stiffness matrix, as indicated in
Equation B5. The different unit geometric matrices introduced in Appendix C can be determined using a form of Equation B5, where the
axial load $N$ is replaced by the unit inertial or external load. Since the beam is directed along the $z$ direction, we focus on the terms where
the loads act in the $z$ direction, all other terms being zero or negligible. The $ij$-component of the matrix $\mathbf{K}_{gt,z}$ is obtained by considering a
unit vertical acceleration:
$$K_{gt,z,ij} = \int_0^L N(z)\frac{d\mathbf{\Phi}_i}{dz} \cdot \frac{d\mathbf{\Phi}_j}{dz}dz, \qquad N(z) = \int_z^L m(z)dz \tag{C4}$$
We write $z_k$ the coordinate of node $k$ along the beam. The $ij$-component of the matrix $\mathbf{K}_{gF,z}^k$ is obtained as:
$$K_{gF,z,ij}^k = \int_0^L N(z)\frac{d\mathbf{\Phi}_i}{dz} \cdot \frac{d\mathbf{\Phi}_j}{dz}dz, \qquad N(z) = 1 \text{ if } z < z_k, 0 \text{ otherwise} \tag{C5}$$
The $ij$-component of the matrix $\mathbf{K}_{g\omega,\alpha\beta}$ is obtained by considering unit centrifugal loads generated using independent rotations around the
unit vectors $\mathbf{e}_x$, $\mathbf{e}_y$, and $\mathbf{e}_z$:
$$K_{g\omega,\alpha\beta,ij} = \int_0^L -\mathbf{e}_z \cdot (\tilde{\mathbf{e}}_\alpha \tilde{\mathbf{e}}_\beta \mathbf{N}(z))\frac{d\mathbf{\Phi}_i}{dz} \cdot \frac{d\mathbf{\Phi}_j}{dz}dz, \qquad \mathbf{N}(z) = \int_z^L m(z)\mathbf{s}_{P_0}dz \tag{C6}$$
Similarly, the $ij$-component of the matrix $\mathbf{K}_{gr,\alpha}$ is:
$$K_{gr,\alpha,ij} = \int_0^L -\mathbf{e}_z \cdot (\tilde{\mathbf{e}}_\alpha \mathbf{N}(z))\frac{d\mathbf{\Phi}_i}{dz} \cdot \frac{d\mathbf{\Phi}_j}{dz}dz, \qquad \mathbf{N}(z) = \int_z^L m(z)\mathbf{s}_{P_0}dz \tag{C7}$$
The Coriolis force, $2m(z)\boldsymbol{\omega} \times (\sum_j \mathbf{\Phi}_j(z)\dot{q}_{e,j})$, can also have an axial contribution. Using Equation C3, the generalize force is:
$$k_{\sigma,\text{Cor},i} = \int_0^L N(z)\mathbf{\Phi}_i \cdot \left[\sum_j \frac{d\mathbf{\Phi}_j}{dz}q_j\right]dz, \qquad N(z) = 2\int_z^L m(z)\sum_k [\omega_x \Phi_{k,y}(z) - \omega_y \Phi_{k,x}(z)]\dot{q}_k \, dz \tag{C8}$$
Equation C8 may be rearranged by introducing a three-dimensional tensor made of shape integrals that are independent of the elastic degrees
of freedom and the rotational speed, and therefore speedup the evaluation of this expression at each time step. The vector $\mathbf{k}_{\sigma,\text{Cor}}(\mathbf{q}_e, \dot{\mathbf{q}}_e)$ is
added to the right hand side of Equation C1.

## C3    Integration into the equations of motion


The term $\mathbf{k}_\sigma = \mathbf{K}_g\mathbf{q}_e$ appears on the third block-row of the equations of motion of the flexible body (Equation A1). Because of the linearity
with respect to the acceleration, rotational velocities, and forces, the different contributions can optionally be incorporated into the third
block-row of the mass matrix ($\mathbf{M}_{e*}$), the term $\mathbf{k}_{\omega,e}$, and the term $\mathbf{f}_e$, respectively. For instance, the term $\sum a_\alpha \mathbf{K}_{gt,\alpha}\mathbf{q}_e$ can be reorganized
as $[\mathbf{K}_{gt}]\mathbf{q}_e \cdot \mathbf{a}$ (using loose notations); therefore, the mass matrix can be updated such that $\mathbf{M}_{xe}$ becomes $\mathbf{M}_{xe} + [\mathbf{K}_{gt}]\mathbf{q}_e$. When a Taylor
expansion is used, such integration is easily implemented as a first-order term (see Appendix D3).

## Appendix D: Alternative formulations

Different formulations of flexible multibody dynamics using shape functions are found in the literature. Some of the alternatives are briefly discussed in this section.

### D1 Jacobian and velocity transformation matrix

In Equation 7, the Jacobian terms $\boldsymbol{J}$ and the virtual work are expressed in vector form. In such form, there is no need to state in which coordinate system the different vectors are expressed. This is convenient to reduce the size of the expressions when using symbolic calculations. In a numerical framework, the vector will have to be expressed in a common frame. When such an approach is used (see, e.g., Lemmer (2018); Branlard (2019)), the Jacobians are sometimes stacked into a matrix form:

$$\boldsymbol{J} = \begin{bmatrix} \boldsymbol{J}_v \\ \boldsymbol{J}_\omega \\ \boldsymbol{J}_e \end{bmatrix} \tag{D1}$$

Some implementation choices are needed depending if these matrices are expressed in the global frame or a body frame. The Jacobian matrices are referred to as "velocity transformation matrix," and the link between formulations in global and local coordinates is given in Branlard (2019). In the same reference, recursive relationships are given for tree-like assembly of bodies to help express the Jacobian matrices of each body recursively, based on the matrices of the parent body. It is also noted that the quadratic velocity terms, $\boldsymbol{k}_\omega$, can be obtained using the time derivative of the Jacobian matrix.

### D2 Rotations and torsion

In this article, we have not elaborated on the change of orientation introduced by shape functions. In most applications, bodies are connected at their extremities and the deflection slope at a body extremity will induce a rotation of the subsequent body (e.g., tilting and rolling of the nacelle at the tower top). The deflection slope can be obtained form the knowledge of the shape functions. This is readily accounted for by introducing a time-varying rotation matrix between bodies, and this is the approach used in our symbolic framework. A formalism of rotations of bodies connected at their extremities is given in Branlard (2019). A more general formulation, introducing shape function rotations $\boldsymbol{\Psi}$, is given in (Wallrapp, 1994; Schwertassek and Wallrapp, 1999; Lemmer, 2018). In such a formulation, the linear rotation field is obtained as $\boldsymbol{I} + \widetilde{\boldsymbol{\Psi}\boldsymbol{q}}$, where $\boldsymbol{I}$ is the identity matrix.

### D3 Shape integrals and Taylor expansion

The results presented in Appendix A consist of integrals over the displaced points of the structure, $\boldsymbol{s}_P = \boldsymbol{s}_{P_0} + \boldsymbol{u}$, where the displacement field is $\boldsymbol{u} = \boldsymbol{\Phi}\boldsymbol{q}_e$. The undeflected position of the structure ($\boldsymbol{s}_{P_0}$) is constant, and the shape functions are known at the initialization; the only time-varying terms are the degrees of freedom $\boldsymbol{q}_e$. Therefore, the integrals can be precomputed by decomposing them into a constant part and a part that is linear with respect to the degrees of freedom $\boldsymbol{q}_e$. The precomputed integrals are referred to as "shape integrals." For a given term $\boldsymbol{T}$ (standing, for instance, for $\boldsymbol{M}_{\theta,\theta}$, $\boldsymbol{C}_t$, $\boldsymbol{C}_r$, $\boldsymbol{G}_r$, $\boldsymbol{G}_e$, or $\boldsymbol{O}_e$), the shape integral expansion is:

$$\boldsymbol{T}(\boldsymbol{q}_e) = \boldsymbol{T}^0 + \sum_{j=1..n_e} \boldsymbol{T}_j^1 q_{e,j} \tag{D2}$$

If $\boldsymbol{T}$ is an array, $\boldsymbol{T}^0$ and $\boldsymbol{T}_j^1$ have the same shape as $\boldsymbol{T}$. As an example, the application of the shape integral expansion to the term $\boldsymbol{M}_{x\theta}$ (see
Equation A3) gives:

$$M_{x\theta} = -\int \tilde{\boldsymbol{s}}_P \mathrm{d}m = \boldsymbol{M}_{x\theta}^0 + \sum_{j=1..n_e} \boldsymbol{M}_{x\theta,j}^1 q_{e,j} \tag{D3}$$

with

$$M_{x\theta}^0 = -\int \tilde{\boldsymbol{s}}_{P_0} \mathrm{d}m, \qquad \boldsymbol{M}_{x\theta,j}^1 = -\int \tilde{\boldsymbol{\Phi}}_j \mathrm{d}m \tag{D4}$$

The zeroth- and first-order shape integrals always consist of integrals over the components of $\boldsymbol{s}_{P_0}$ and $\boldsymbol{\Phi}$, which can be precomputed for a
given flexible body. We note that the precomputed shape integrals can in turn be obtained from intermediate integrals (e.g., the $S_*$ and $N_*$
terms introduced by Wallrapp (Wallrapp, 1994), or the $\sigma$, $\Sigma$, $\Upsilon$, $\Psi$ terms introduced by Shabana (Shabana, 2013)). The zeroth- and first-order
shape integrals are stored using a "Taylor" object-oriented class in the standard input data format defined by Wallrapp. The YAMS library
can compute the shape integrals using a direct integration or using a finite-element formulation (see Schwertassek and Wallrapp (1999)).
The geometric stiffness introduced in Appendix C is linear in the elastic degrees of freedom $\boldsymbol{q}_e$. Therefore, the unit geometric stiffness
matrices (which are also shape integrals) can be conveniently added into the first-order terms of Equation D2. For instance, if we write $\boldsymbol{M}_{ex}$
(given in Equation A6) using a first-order expansion, $\boldsymbol{M}_{ex} = \boldsymbol{M}_{ex}^0 + \boldsymbol{M}_{ex}^1 \boldsymbol{q}_e$, then the geometric stiffening effect can directly be inserted
into the first-order term, such that $\boldsymbol{M}_{ex}^1$ becomes $\boldsymbol{M}_{ex}^1 + \boldsymbol{K}_{gt}$. Similarly, the term $\boldsymbol{K}_{gr}$ can be inserted into $\boldsymbol{M}_{\theta e}^1$, $\boldsymbol{K}_{g\omega}$ into $\boldsymbol{O}_e^1$, $\boldsymbol{K}_{gF}$ into
$\boldsymbol{\Phi}^1$, and $\boldsymbol{K}_{g\tau}$ into $\boldsymbol{\Psi}^1$ in the calculation of the generalized forces. The different contributions are summarized in Table 6.9 of the book of
Schwertassek and Wallrapp (1999). A shortcoming of inserting the geometric stiffness effects into the first-order coefficient is that it could
make the mass matrix symmetric (if the user code assumes $\boldsymbol{M}_{xe} = \boldsymbol{M}_{ex}^t$), instead of acting only on the third block-row of the mass matrix.

## 669 D4   Taylor expansion of the displacement field

In the work of Wallrap (Wallrapp, 1993, 1994), the displacement field is assumed to be a function of the degrees of freedom, $\boldsymbol{u} = \boldsymbol{\Phi}_u(\boldsymbol{q}_e)\boldsymbol{q}_e$,
where $\boldsymbol{\Phi}_u$ consists of a Taylor series expansion of the shape functions that contain $\Phi^0$ and $\Phi^1$ terms. The resulting equations of motion are
still expressed using shape integrals of the form given in Equation D2, but the 1 terms will contain some additional integrals over $\Phi^1$. The
advantage of this method is that the $\Phi^1$ terms effectively account for the geometric stiffness. In practice, it is equivalent, and as convenient,
to neglect the $\Phi^1$ terms and introduce the geometric stiffness using the method presented in Appendix C (and optionally integrate them into
the 1 terms as presented in Appendix D3).

## 676 D5   ElastoDyn and the partial loads approach

The ElastoDyn module of OpenFAST (Jonkman et al., 2021) uses the so-called "partial loads" approach to implement the equations of
motion. The underlying theory used to derive the equations of motion is the same as Kane's formalism presented in section 2, but the partial
load approach takes advantage of the fact that the calculation of reaction loads or point loads at body extremities requires similar terms to the
ones needed for the equations of motion. In the discussion below, we assume that the different bodies of the structure form a tree structure
with the root at the bottom and the leaves above. For a tree-like structure, there is a natural relationship between loads in the structure and the
degrees of freedom. A virtual displacement of a given degree of freedom will only displace the structure above it. The equation of motion
of this degree of freedom can therefore be obtained from the virtual work of the loads at a point located just above the degree of freedom,
as if the entire structure above was replaced by lumped loads. The point loads contain contributions from the external loads above the point
in consideration, but also inertial and gyroscopic loads associated with all the degrees of freedom of the system. If the point is at a joint, the
loads corresponds to the reaction loads at this point. We write $P$ the point located after a given degree of freedom $r$. The equation of motion
for this degree of freedom is obtained as if the system was isolated:
$$\mathrm{f}_r + \mathrm{f}_r^* = 0 = \boldsymbol{J}_{v_P,r} \cdot \boldsymbol{f}_P + \boldsymbol{J}_{\omega_P,r} \cdot \boldsymbol{\tau}_P + h_r \tag{D5}$$
where: $\boldsymbol{J}_{v_P,r}$ and $\boldsymbol{J}_{\omega_P,r}$ are the partial velocities of point $P$ with respect to the degree of freedom $r$; $\boldsymbol{f}_P$ and $\boldsymbol{\tau}_P$ are 3-vectors containing the
force and torque from the structure above the degree of freedom $r$ (including external and inertial contributions); and $h_r$ is the generalized
load associated with the isolated degree of freedom $r$ (e.g., the elastic loads for a flexible body, or the spring and damping loads for a degree
of freedom representing a joint). The point loads $\boldsymbol{f}_P$ and $\boldsymbol{\tau}_P$ can be decomposed into terms that are proportional to the accelerations of all
the degrees of freedom (indexed with $r$) and additional terms (labeled "$t$"):
$$\boldsymbol{f}_P = \sum_{j=1}^{n_q} \boldsymbol{f}_{P,j} \ddot{q}_j + \boldsymbol{f}_{P,t}, \qquad \boldsymbol{\tau}_P = \sum_{j=1}^{n_q} \boldsymbol{\tau}_{P,j} \ddot{q}_j + \boldsymbol{\tau}_{P,t} \tag{D6}$$
The terms $\boldsymbol{f}_{P,r}$ and $\boldsymbol{\tau}_{P,r}$ act as generalized masses and they are referred to as "partial loads". Combining Equation D5 and Equation D6, the
term $rj$ of the mass matrix and the term $r$ of the right hand side of the equation of motion (Equation 22) are obtained as:
$$M_{rj} = -\boldsymbol{J}_{v_P,r} \cdot \boldsymbol{f}_{P,j} - \boldsymbol{J}_{\omega_P,r} \cdot \boldsymbol{\tau}_{P,j}, \qquad F_r = \boldsymbol{J}_{v_P,r} \cdot \boldsymbol{f}_{P,t} + \boldsymbol{J}_{\omega_P,r} \cdot \boldsymbol{\tau}_{P,t} + h_r \tag{D7}$$
Therefore, the knowledge of the partial loads and the partial velocities at key points of the structure (typically, points where user outputs
are desired) can be used to obtain the reaction loads (Equation D6) and the equations of motion (Equation D7). This is the approach used
in ElastoDyn: the loads at key points of the structure were derived using hand calculations, and then the partial loads were used for the
implementation of the outputs and the equations of motion. The reader is referred to the notes provided in the online documentation of
ElastoDyn for more details (Jonkman et al., 2021). A general procedure to obtain partial loads can be devised (using kinematics to find
velocities and acceleration in the structure, and computing the loads from the tree top to the root), but would be beyond the scope of this
article.

## Appendix E:  Equations of motion of simple wind turbine models

In this section, we present the equations of motion for the examples presented in section 4.

### E1   Two-degrees-of-freedom model of a land-based or fixed-bottom wind turbine

In this section, we provide some intermediate values to obtain the equations of motion given in subsection 4.4. We use the hat notation to
indicate unit vectors of a frame, where the frame is identified as $t$, $n$, $r$ for the tower, nacelle, and rotor, respectively. For instance, $v\hat{t}_x$ is the
unit vector in the $x$ direction of the tower frame. The degrees of freedom are $\boldsymbol{q} = (q, \psi)$. The kinematics of the tower (at its origin) are zero:
$$\boldsymbol{v}_{O,T} = \boldsymbol{0}, \quad \boldsymbol{\omega}_T = \boldsymbol{0}, \quad \boldsymbol{a}_{O,T} = \boldsymbol{0} \tag{E1}$$
All Jacobians are zero except $\boldsymbol{J}_{e,1T} = 1$ The inertial force, torque, and elastic force are:
$$\boldsymbol{f}_T^* = C_{tTx} \ddot{q} \hat{\boldsymbol{t}}_x + M_T g \hat{\boldsymbol{t}}_z, \quad \boldsymbol{\tau}_T^* = C_{rTy} \ddot{q} \hat{\boldsymbol{t}}_y, \quad \boldsymbol{E}_T^* = f_e + D_e \dot{q} + (K_e + K_q)q + M_e \ddot{q} \tag{E2}$$
The nacelle kinematics (at its center of mass) are:
$\boldsymbol{v}_{G,N} = \dot{q}\hat{\boldsymbol{t}}_x + \nu_y z_{NG}\dot{q}\hat{\boldsymbol{n}}_x - \nu_y x_{NG}\dot{q}\hat{\boldsymbol{n}}_z, \quad \boldsymbol{\omega}_N = \nu_y \dot{q}\hat{\boldsymbol{t}}_y$     (E3)
$\boldsymbol{a}_{G,N} = \ddot{q}\hat{\boldsymbol{t}}_x + \left(-\nu_y^2 x_{NG}\dot{q}^2 + \nu_y z_{NG}\ddot{q}\right)\hat{\boldsymbol{n}}_x + \left(-\nu_y^2 z_{NG}\dot{q}^2 - \nu_y x_{NG}\ddot{q}\right)\hat{\boldsymbol{n}}_z$     (E4)
The Jacobians with respect to $q$ are:
$\boldsymbol{J}_{v,1N} = \hat{\boldsymbol{t}}_x + \nu_y z_{NG}\hat{\boldsymbol{n}}_x - \nu_y x_{NG}\hat{\boldsymbol{n}}_z, \quad \boldsymbol{J}_{\omega,1N} = \nu_y \hat{\boldsymbol{t}}_y$     (E5)
The inertial force and torque on the nacelle are:
$\boldsymbol{f}_N^* = M_N \ddot{q}\hat{\boldsymbol{t}}_x + M_N \left(-\nu_y^2 x_{NG}\dot{q}^2 + \nu_y z_{NG}\ddot{q}\right)\hat{\boldsymbol{n}}_x + M_N \left(-\nu_y^2 z_{NG}\dot{q}^2 - \nu_y x_{NG}\ddot{q}\right)\hat{\boldsymbol{n}}_z, \quad \boldsymbol{\tau}_N^* = J_{y,N}\nu_y \ddot{q}\hat{\boldsymbol{n}}_y$     (E6)
The kinematics of the rotor are:
$\boldsymbol{v}_{G,R} = \dot{q}\hat{\boldsymbol{t}}_x + \nu_y z_{NR}\dot{q}\hat{\boldsymbol{n}}_x - \nu_y x_{NR}\dot{q}\hat{\boldsymbol{n}}_z, \quad \boldsymbol{\omega}_R = \dot{\psi}\hat{\boldsymbol{e}}_{\mathbf{rx}} + \nu_y \dot{q}\hat{\boldsymbol{t}}_y$     (E7)
$\boldsymbol{a}_{G,R} = \ddot{q}\hat{\boldsymbol{t}}_x + \left(-\nu_y^2 x_{NR}\dot{q}^2 + \nu_y z_{NR}\ddot{q}\right)\hat{\boldsymbol{n}}_x + \left(-\nu_y^2 z_{NR}\dot{q}^2 - \nu_y x_{NR}\ddot{q}\right)\hat{\boldsymbol{n}}_z$     (E8)
The corresponding Jacobians with respect to $q$ ("1") and $\psi$ ("2") are:
$\boldsymbol{J}_{v,1R} = \hat{\boldsymbol{t}}_x + \nu_y z_{NR}\hat{\boldsymbol{n}}_x - \nu_y x_{NR}\hat{\boldsymbol{n}}_z, \quad \boldsymbol{J}_{\omega,1R} = \nu_y \hat{\boldsymbol{t}}_y, \quad \boldsymbol{J}_{\omega,2R} = \hat{\boldsymbol{r}}_x$     (E9... )
The inertial force and torque on the rotor are:
$\boldsymbol{f}_R^* = M_R \ddot{q}\hat{\boldsymbol{t}}_x + M_R \left(-\nu_y^2 x_{NR}\dot{q}^2 + \nu_y z_{NR}\ddot{q}\right)\hat{\boldsymbol{n}}_x + M_R \left(-\nu_y^2 z_{NR}\dot{q}^2 - \nu_y x_{NR}\ddot{q}\right)\hat{\boldsymbol{n}}_z$     (E9)
$\boldsymbol{\tau}_R^* = J_{x,R}\ddot{\psi}\hat{\boldsymbol{r}}_x$     (E10)
$+ \left(J_{\oplus,R}\nu_y \sin\left(\psi\right)\dot{\psi}\dot{q} + J_{\oplus,R}\left(-\nu_y \sin\left(\psi\right)\dot{\psi}\dot{q} + \nu_y \cos\left(\psi\right)\ddot{q}\right) - J_{x,R}\nu_y \sin\left(\psi\right)\dot{\psi}\dot{q}\right)\hat{\boldsymbol{r}}_y$     (E11)
$+ \left(J_{\oplus,R}\nu_y \cos\left(\psi\right)\dot{\psi}\dot{q} + J_{\oplus,R}\left(-\nu_y \sin\left(\psi\right)\ddot{q} - \nu_y \cos\left(\psi\right)\dot{\psi}\dot{q}\right) - J_{x,R}\nu_y \cos\left(\psi\right)\dot{\psi}\dot{q}\right)\hat{\boldsymbol{r}}_z$     (E12)
**E2   Three-degrees-of-freedom model of a land-based or fixed-bottom wind turbine**
The equations of motion for the model presented in subsection 4.5, with $\boldsymbol{q} = (q_1, q_2, \psi)$, are given in this section. The elements of the mass
matrix are:
$M_{11} = \left[M_{e11} + M_N + M_R\right]$     (E13)
$+ \left[J_{y,N} + J_{\oplus,R} + M_N \left(x_{NG}^2 - 2x_{NG}q_1 + z_{NG}^2\right) + M_R \left(x_{NR}^2 - 2x_{NR}q_1 + z_{NR}^2\right)\right]\nu_y^2$     (E14)
$+ 2\left[M_N z_{NG} + M_R z_{NR}\right]\nu_y$     (E15)
$M_{13} = J_{x,R}\theta_t \nu_x \nu_y q_2$     (E16)
$M_{22} = \left[M_{e22} + M_N + M_R\right]$     (E17)
$+ \left[J_{x,N} + J_{x,R} + M_N z_{NG}^2 + M_R z_{NR}^2\right]\nu_x^2$     (E18)
$- 2\left[M_N z_{NG} + M_R z_{NR}\right]\nu_x$     (E19)
$M_{23} = J_{x,R}\nu_x$     (E20)
$M_{33} = J_{x,R}$     (E21)

The elements of the forcing vector are:

$$f_1 = f_{e1} - K_{e11}q_1 - D_{e11}\dot{q}_1 - J_{x,R}\theta_t\nu_x\nu_y\dot{\psi}\dot{q}_2 + [M_N x_{NG} + M_R x_{NR}]\nu_y^2\dot{q}_1^2 \tag{E22}$$

$$+ g\left[M_N\left(\nu_y^2 z_{NG}q_1 + \nu_y x_{NG}\right) + M_R\left(\nu_y^2 z_{NR}q_1 + \nu_y x_{NR}\right)\right] + f_a\left[\theta_t\nu_y x_{NR} - \theta_t\nu_y q_1 + \nu_y z_{NR} + 1\right] \tag{E23}$$

$$f_2 = f_{e2} - K_{e22}q_2 - D_{e22}\dot{q}_2 + J_{x,R}\theta_t\nu_x\nu_y\dot{\psi}\dot{q}_1 \tag{E24}$$

$$+ g\left[M_N z_{NG} + M_R z_{NR}\right]\nu_x^2 q_2 + f_a\theta_t\nu_x q_2 \tag{E25}$$

$$f_3 = -J_{x,R}\theta_t\nu_x\nu_y\dot{q}_1\dot{q}_2 + \tau_a \tag{E26}$$

## E3 Three-degrees-of-freedom model of a floating wind turbine

The equations of motion for the model presented in subsection 4.6, with $\boldsymbol{q} = (x, \phi, q_T)$, are given in this section. The elements of the mass matrix are:

$$M_{11} = M_F + M_T + M_N \tag{E27}$$

$$M_{12} = M_F z_{FG} - M_{dTz} + M_N\left[L_T + z_{NG} - \nu_y x_{NG}q_T - \phi_y(x_{NG} + q_T + \nu_y z_{NG}q_T)\right] \tag{E28}$$

$$M_{13} = C_{tT1x} + M_N\left[1 + \nu_y z_{NG} - \nu_y^2 x_{NG}q_T - \phi_y(\nu_y^2 z_{NG}q_T + \nu_y x_{NG})\right] \tag{E29}$$

$$M_{22} = J_{y,F} + M_F z_{FG}^2 + J_{T,y} + J_{y,N} + M_N\left[(L_T^2 + z_{NG})^2 + (q_T + x_{NG})^2 + 2\nu_y q_T(z_{NG}q_T - L_T x_{NG})\right] \tag{E30}$$

$$M_{23} = C_{rT1y} + \left[J_{y,N} + M_N(x_{NG}^2 + z_{NG}^2 + L_T z_{NG} + \nu_y q_T(z_{NG}q_T - L_T x_{NG})\right]\nu_y + M_N\left[L_T + z_{NG}\right] \tag{E31}$$

$$M_{33} = M_e + M_N + \left[J_{y,N} + M_N\left(x_{NG}^2 - 2x_{NG}q_T + z_{NG}^2\right)\right]\nu_y^2 + 2M_N\nu_y z_{NG} \tag{E32}$$

The elements of the forcing vector are:

$$f_1 = f_H + \left[M_F z_{FG} - M_{dz} + M_N(L_T + z_{NG} - \nu_y x_{NG}q_T)\right]\phi_y\dot{\phi}_y^2 + M_N\left[q_T + x_{NG} + \nu_y z_{NG}q_T\right]\dot{\phi}_y^2 \tag{E33}$$

$$+ \left[2C_{tx} + M_N(1 + \nu_y z_{NG} - \nu_y^2 x_{NG}q_T)\right]\phi_y\dot{\phi}_y\dot{q}_T + M_N\nu_y\left[x_{NG} + \nu_y z_{NG}q_T\right]\dot{\phi}_y\dot{q}_T \tag{E34}$$

$$+ M_N\nu_y^2\left[x_{NG} + z_{NG}\phi_y\right]\dot{q}_T^2 \tag{E35}$$

$$+ f_a\left[1 - \theta_t\nu_y q_T - \nu_y\phi_y q_T\right] \tag{E36}$$

$$f_2 = \tau_H + M_N\left[\nu_y^2(L_T x_{NG} - z_{NG}q_T)\right]\dot{q}_T^2 \tag{E37}$$

$$- 2M_N\left[q_T + x_{NG} + \nu_y(2z_{NG}q_T - L_T x_{NG}) - \nu_y^2 q_T(L_T z_{NG} + x_{NG}q_T)\right]\dot{\phi}_y\dot{q}_T \tag{E38}$$

$$+ g\left[M_F z_{FG}\phi_y - M_{dz}\phi_y + M_N\left\{(L_T + z_{NG} - \nu_y x_{NG}q_T)\phi_y + q_T + x_{NG} + \nu_y z_{NG}q_T\right\}\right] \tag{E39}$$

$$+ f_a\left[L_T + z_{NR} + \theta_t x_{NR} + \theta_t q_T + \nu_y q_T^2 - L_T\theta_t\nu_y q_T\right] \tag{E40}$$

$$f_3 = f_e - D_e\dot{q}_T - K_e q_T \tag{E41}$$

$$+ M_N\left[q_T + x_{NG} + \nu_y(2z_{NG}q_T - L_T x_{NG}) - \nu_y^2 q_T(L_T z_{NG} + x_{NG}q_T)\right]\dot{\phi}_y^2 \tag{E42}$$

$$+ M_N\nu_y^2 x_{NG}\dot{q}_T^2 \tag{E43}$$

$$+ g\left[C_{tT1x}\phi_y + M_N\left(\nu_y x_{NG} + \nu_y^2 z_{NG}q_T - \nu_y^2 x_{NG}\phi_y q_T + \nu_y z_{NG}\phi_y + \phi_y\right)\right] \tag{E44}$$

$$+ f_a\left[1 + \theta_t\nu_y x_{NR} - \theta_t\nu_y q_T + \nu_y z_{NR}\right] \tag{E45}$$

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
