# Peer review of "A symbolic framework for flexible multibody systems applied to horizontal-axis wind turbines"

_Wind Energy Science, 2021_

## Author Comment (AC1)

Dear reviewers

Thank you so much for your time and effort in reviewing our paper. Please find our answers to your comments below (in blue). If given a chance, we will submit a revised manuscript together with a pdf highlighting the differences made to the text.

Emmanuel and Jens

**Reviewer 1**

After reviewing carefully, the paper titled "A symbolic framework for flexible multibody system applied to horizontal axis wind turbines", I have arrived to the following conclusion: Without underestimating the programming effort and authors' particular motivation, I strongly think that the proposed work does not contribute in any form to the current state of the art in flexible multibody dynamics applied to horizontal axis wind turbines. The methods proposed are not new. Moreover, the assumption made on the elastic deformation of the flexible bodies is quite limited and thus, it requires amendments such us the inclusion of softening/stiffening terms. There are some formulations available in the literature, for instance the geometrically exact beam developed during the eighties and nineties, that can be already implemented in "pseudo" real time and are by far superior to the approach proposed by the authors. Therefore, based on my assessment I regrettably cannot recommend the paper for publication, even after major changes.

> We would like to thank the reviewer for their time in reviewing our paper. Your comments are useful for us, as it indicates that we need to stress further the relevance of this work and how it stands out. We provide different points below to clarify our vision and line of thoughts. In the revised version of the manuscript, we will include changes to highlight most of these points.

- The paper does not intend to contribute directly to the state of the art in flexible multibody dynamics applied to horizontal axis wind turbine. As you noted, and as we mention in the text, the theory is not new. In the revised version, we will highlight the fact that it does not correspond to the state of the art. Geometrically exact beam theory and general multibody frameworks exist and they are in many aspects superior to the framework presented here. We would argue though that they target different applications than the ones we foresee for our work, which are: computationally efficient and simple non-linear and linear models. We discuss potential applications of our framework in section 5.1 of the paper (e.g. frequency-domain analysis, stability analysis, control design and physics-based digital twins), and we believe that more advanced tools cannot be used for such applications, or are less convenient. In our opinion, the field of control engineering is one which can benefit the most from having access to simple linear and non-linear models that are physics-based and generated in a mostly automated fashion. This is an innovation in the field of control engineering, where most models are still based on heuristic assumptions and adhoc methods or obtained from numerical linearizations of high-fidelity models which are hard to relate to the underlying physics. One of the contributions from this paper is therefore the re-appreciation of known methods so that they can be applied systematically in new applications. In the revised version of the paper, will we stress the distinction between our approach and state-of-the-art methods early in the introduction.

- Part of the motivation behind this work is to provide a reference for the underlying theory behind tools such as Flex and OpenFAST, which also use a shape function approach and the concept of virtual velocities. Neither Flex nor the ElastoDyn module of OpenFAST have an official publication detailing their theory, therefore we believe that our development is relevant for people using these tools and wanting to develop them further. We can support this need from our experience exchanging with users and developers of these tools in academia and in the industry: flex-based tools are still actively used by the industry, many question and confusion arise for the users, and the authors have provided support for projects needing further developments of these tools.
- As you pointed out, the theory is not new, but we believe that our presentation of the theory is unique in the fact that it is concise. It also covers rigid and flexible bodies within the same context. We point multiple times to the reference books of Shabana and Schwertassek and Wallrapp, which cover the theory in great details, though the latter is mostly available to the German-speaking audience. It took us many years to grasp some of the developments present in these books, and to try to bridge the gap between them and the practical implementations of tools such as Flex and ElastoDyn. We therefore believe that our concise version will be relevant to the reader curious about applying this theory and can be an accelerator before diving into these textbooks.
- The topic of geometrical stiffening is indeed an important topic. In the first version of this paper, we chose to only address it via two examples which are likely the most relevant for wind turbines: the centrifugal stiffening and the tower softening. In the revised version of the paper, we are adding a new section dedicated to geometrical stiffening. We present general equations of the geometrical stiffening (as function of accelerations, rotational velocity products and loads) and we present how the different terms are computed using the shape function approach. We point the reader to the book of a Schwertassek and Wallrapp for the computation of these terms using a finite-element approach. We are grateful that your review incited us to add this section to the manuscript.
- One of the main contribution and novelty behind our work is the fact that the equations of motions are obtained in a symbolic manner (using a formal/symbolic calculator, also called computer algebra system), instead of numerically. The theory section presents what is implemented behind the scene in our symbolic calculator to systematically determine the equations of motion and linearize them if needed. Once the equations of motion are determined symbolically, they can be printed to screen, converted to latex, or converted to software code, including optimizations to eliminate repeated terms. Most importantly, the equations can be linearized analytically. We discuss the advantage and inconvenient of the symbolic framework in section 5.2 and 5.4. The framework indeed represents a major programming effort, as it not only consists in the symbolic framework but also pre and processing tools to compute shape integrals and structural parameters of a given structure using either a finite-element method or a shape function approach (and export it in the "standard-input-data" format). We believe it is relevant to the community as the tool and preprocessors are distributed as an open-source library. In the revised version of the manuscript, we will slightly expand on the implementation effort in section 3.
- We are confident that our work has many applications (as listed in section 5.1), for research, production, and education. We believe that the list of applications also justifies our work and the publication of this manuscript. We have developed this framework to answer our research needs, with the goal of making it available to the community. Despite the advance in different fields (aerodynamics, structure, control, etc.), and the many experts in each area, multidisciplinary

applications are not straightforward and they typically require simpler and faster models. The examples presented in the article are simple, but they can be used both for research and educational purposes, they present some of the key elements of wind turbine dynamics and their expressions cannot be found in this level of details in the literature. The framework can of course be used to generate equations of motions of systems more complex than the ones presented in this work. In this aspect, the authors are already using the framework to generate linear models of structures and use them in research applications, such as controller tuning, frequency-domain analyses, and digital twinning. We will briefly extend on the possible applications in the revised version to further highlight the relevance of this work.

The points above present some of the novelties of our work, we hope that they can help understand our line of thoughts and why we believe that our work is relevant as a research output to the community. We have taken your comments in consideration for the revised version of the manuscript to further clarify some of the points above.

**Reviewer 2**

As the title states, this paper presents a symbolic framework for flexible multibody systems applied to HAWTs. Overall, the approach is clearly presented and shows excellent agreement to OpenFAST. Moreover, in addition to the paper, a companion open-source Python implementation is provided so readers can repeat the analysis and apply the approach to general systems. In contrast to the other reviewers comments, I would strongly encourage publication of this article. Novel items include a (1) clear and concise presentation of flexible multibody dynamics expressed in Kane's formulation, applicable to both nonlinear and linearized systems, in symbolic form and (2) an open-source Python implementation based on SymPy and PyDy. While the formulation will not replace widely used structural dynamics software for HAWTs such as OpenFAST, Bladed, HAWC2, and FLEX5, that is not the intent, as stated. While the approach could be used to simplify or enhance the ElastoDyn module of OpenFAST, more important applications, as stated, include frequency-domain analysis important in preliminary design, stability analysis, controls design, physics-based digital twins, etc. The approach does account for centrifugal stiffenning and gravitational destiffenning.

> Thank you for your review.

Please a find a few specific comments below:

Section 2: The ElastoDyn module of OpenFAST uses the concept of "partial loads", which is extension of the "partial velocity" approach used by Kane's method. Partial loads simplify the formulation of the equations of motion into terms that are useful for load output calculations once the equations of motion are solved. I don't see this concept mention, but perhaps it would be an interesting extension of the described approach, if possible?

> Thank you for pointing this out. In the revised version of the manuscript, we will mention the partial loads approach in Appendix C to bridge the gap between the different formulations.

Page 6, Line 141: It would be useful to mention the contribution of the inertia term (M*qdd) on the stiffness matrix, which can impact the linearized solution if the model is not in steady state, where otherwise qdd = 0).

> You are correct. The mass matrix depends on q and therefore a stiffening term can occur if qdd0 is non-zero. For completeness, we have modified the equation to add qdd0 into the linearized stiffness matrix and we have added the following to the text: "In practical applications, linearization is done at an operating point where the acceleration is zero (qddot_0=0) and most velocities in qdot are also zero."

Page 6, Line 158: "Shape functions of any order" are mentioned. Is this referring to shape functions expressed as polynomials? Are other forms of shape functions permitted in the Python implementation?

> The order of the shape function in this context refers to the order of the Taylor series used to describe the flexible deformation depending on the elastic coordinates q_e. The shape function itself is not limited to a polynomial representation. We have slightly modified the text and pointed the user to the section where the expansion is referenced: "allowing the symbolic computation with Taylor expansions to any order. In practice, a zeroth or 1st order expansion is used. The use of Taylor expansion is presented in Appendix C." We will also rewrite appendix C to clarify the Taylor expansion formalism (for instance, the first order expansion of the shape function is mainly used for the geometrical stiffening effects).

**Reviewer 3**

This work provides a fairly general description of the ODEs that describe multibody dynamics and then explains how they can be implemented in a Python library (YAMS) for symbolic manipulations, using it to solve several problems in the context of wind turbines. Overall, it is a rigorous work that does not introduce any theoretical novelties but focuses on the description and validation of a new software library. The theory of multibody analysis is based on Kane's method (for rigid bodies) and Shabanna's (for flexible components), and it is fairly standard, although not based on the most recent and advanced formulations. To make the work self-contained, many details are given in the appendices. The implementation of the Python library is presented succinctly in section 3. Four examples are presented in section 4 illustrating the convenience of the library to solve problems related to the dynamics of wind turbines.

> Thank you for your review

The comparison of the results obtained with the new framework against existing codes validates it.

The transient simulations of section 4 show the evolution of the solution. However, no information is given regarding the time-integration scheme employed by YAMS, but should be provided.

We have added the following statement in the revised version: "The time integration in YAMS currently relies on tools provided in the SciPy package, which implements several time integrators. Sufficient level of accuracy was obtained using a 4th-order Runge-Kutta method which is the default method. Kane's method, which uses a minimal set of coordinates, tends to lead to stiff system, and it is possible that implicit integrators may be needed for other systems."

The library YAMS will be a useful prototyping tool for researchers working with wind turbines. The authors offer it free of charge, including its source code. Since it is programmed in Python, it can be run in all major operating systems also for free. The article lacks theoretical or methodological novelties but presents a tool that can be useful for the scientific community. It is clearly written with meaningful examples and figures are very illustrative. I would recommend its publication with the minor correction alluded to before.

> Thank you again for your time and your review.

---

## Author Response (AR2)

Dear Editor,

Thank you for carefully reviewing our work and for your comments. We have tried to address them, and we believe the paper is now improved thanks to your suggestions. We will look forward to your feedback should you have additional comments, and we will be happy to address them.

Emmanuel and Jens

Now the paper's focus is clearer. However, the title, abstract, introduction and presentation suggest you are proposing a contribution in multibody dynamics, which is not the case. What you are proposing may represent a contribution within control systems. It may have some educational value as well. Both aspects are very important nowadays. Therefore, I recommend to reformulate the title, abstract and introduction of the paper to make transparent the goal and possible contributions.

> Based on your comment we have now reformulated the title to the following: "A symbolic framework to obtain mid-fidelity models of flexible multibody systems with application to horizontal-axis wind turbines"
> We have also modified the abstract and the introduction to stress the fact that our work focuses on symbolic calculation, and mid-fidelity models compared to what is possible with more advanced nonlinear dynamics treatment of the problem.

After reviewing carefully, the revised version of the paper titled "A symbolic framework for flexible multibody system applied to horizontal axis wind turbines", I have the following comments:

• Page 2: Added text is not precise. The key point regarding the geometrically exact beam model (GEBM) is that the kinematics is exactly represented, i.e., R^3 x SO(3) or SE(3). It has nothing to do with nonlinear shape functions. Moreover, the linearization of GEBM is typically computed straightforwardly as well. This is nothing else than a standard procedure in implicit dynamics.
  > Thank you for your comment, we have modified the sentence as follows: "The geometrically exact beam theory is more precise than the shape function approach because it represents the beam kinematics exactly. Linearization of the geometrical exact beam theory equations is possible and also more precise than the shape function approach but it leads to larger and more involved expressions."
• Pages 9, 10 and 11: The geometric stiffness matrix is not complete. Adopting a floating frame of reference and assuming that the beam is initially aligned with the third unit vector of such a floating frame, the terms that are missing are the proportional ones to $v_1 * \omega_2$, $-\omega_1 * v_2$, $- d/dt\, v_3$ and $g_3$. For instance, if the attachment of the blade/tower is moving back-and-forth and side-to-side and under action of gravity, all the missing terms are going to be activated. Whether they are small or large is another question. Those terms need to be accounted for to provide a consistent linearization. Blade or tower, the structure of the geometrical stiffness matrix should be the same.
  > We agree with your statement. We have added the following sentences to the section regarding the blade: "In this example, the beam rotates with respect to a fixed support, the influence of gravity is omitted and no force other than the centrifugal force is assumed in the radial direction (the Coriolis force contribution to the radial force is assumed to be negligible for simplicity). Therefore, the only geometrical stiffening comes from the centrifugal force.

For a wind turbine blade mounted on a flexible support and under the influence of gravity, the different geometrical stiffening terms presented in Appendix C should be used. "

➢ In the tower example, we have added a similar sentence: "The tower is assumed to be fixed and under no significant vertical external loads and therefore the only geometrical stiffness comes from the gravitational force. For a tower mounted on a moving support (fixed-bottom foundation or floater), additional geometrical stiffening terms would be present (see Appendix C)."

➢ The terms - d/dt v_3 and g_3 are accounted for by term K_gt,alpha presented equation C2. In this appendix though, and to your point, we do not have the impact of the Coriolis force on the geometrical stiffening. We believe that in the approach of Schwertassek and Wallrapp, this term was neglected as it was considered to be of second order, though they never explicitly mention the Coriolis force when they discuss geometric stiffening. The term can be considered of second-order because it contains a "q"x"qdot" term (it is a stiffening or damping term). Yet, we agree that this the term might not be insignificant if the steady state deflections are large. We have added some comments and expressions in Appendix C to address this issue.

- Page 18: Claims regarding the linearization are again imprecise or even misleading. Analytical linearization is indeed possible and a usual practice in multibody dynamics. What is not trivial is the reduction of the model due to the fact that the dynamics lives on a nonlinear manifold. I do not see how you can provide different levels of details for instance without considering multiscale analysis in time and space… You should elaborate more on this.

➢ We agree that two slightly different nonlinear systems may lead to completely different responses, and obtaining reduced-order models is not trivial, and it's a topic where we are not well versed. We have added the following sentences to attempt to address the topic: "The shape function approach is an approximate method: it introduces a separation of space and time early on in the development of the nonlinear equations of motion, and applies low order polynomial (usually linear or quadratic) approximations to eliminate high-order terms (see e.g. Table 1 of Wallrapp,1994). This was presented as an advantage in Section 5.1 because the equations are obtained in compact form and are readily linearized. The approximations introduced by the method may imply that nonlinearities are not well captured, which is why the models are labeled as "mid-fidelity" throughout this article. The domain of validity of the nonlinear or linear models presented may therefore be limited in time and space as opposed to fully nonlinear models. Advanced methods to obtain high-fidelity reduced-order models from nonlinear dynamic systems are beyond the scope of this work, see, e.g. Steindl:2001, Benner:2015, Touze:2021."